# Tracing rate and extent of human induced hypoxia during the last 200 years in the mesotrophic lake Tiefer See (NE Germany)

Ido Sirota[1*], Rik Tjallingii[1], Sylvia Pinkerneil[1], Birgit Schroeder[1], Marlen Albert[1], Rebecca Kearny[1], Oliver Heiri[2], Simona Breu[2] and Achim Brauer[1,3]

[1]Section Climate Dynamics and Landscape Evolution, GFZ German Research Centre for Geosciences, Potsdam, Germany

[2]Geoecology, Department of Environmental Sciences, University of Basel, Basel, Switzerland

[3]Institute of Geosciences, University of Potsdam, Potsdam, Germany

*Corresponding author: Ido Sirota (idosir@gfz-potsdam.de)

**Abstract.** The global spread of lake hypoxia, [$O_2$] < 2 mg/l, during the last two centuries has a severe impact on ecological systems and sedimentation processes. While the occurrence of hypoxia was observed in many lakes, a detailed quantification of hypoxia spread at centennial timescales remained largely unquantified. We track the evolution of hypoxia and its controls during the past 200 yrs in lake Tiefer See (TSK; NE Germany) using 17 gravity cores, recovered between 10 and 62 m water depth in combination with lake monitoring data. Lake hypoxia was associated with the onset of varve preservation in the TSK, and has been dated by varve counting to 1918±1 at 62 m water depth and reached a lake-floor depth of 16 m in 1997±1. This indicates oxygen concentration to fell below the threshold for varve preservation at the lakefloor (>16 m). Sediment cores at 10-12 m depth do not contain varves indicating well oxygenation of the upper water column. Monitoring data show that the threshold for hypoxia, intensity and duration of hypoxia which are sufficient for varve preservation, is a period of five months of [$O_2$] < 5 $\frac{mg}{l}$ and two months of [$O_2$] < 2 $\frac{mg}{l}$. Detailed TOC, $\delta^{13}C_{org}$ and XRF core scanning analyses of the short cores indicate that the decline in DO started several decades prior to the varve preservation. This proves a change in the depositional conditions in the lake following a transition phase of several decades during which varves were not preserved. Furthermore, varve preservation does occur at seasonal stratification and not necessarily requires permanent stratification.

**Keywords:** Hypoxia, Lake sediments, Varves, XRF core scanning, environmental monitoring

## 1    Introduction

The vertical distribution of dissolved oxygen (DO) in lakes and shallow marine environments is among the most important factors regulating the ecology, biogenic and chemical conditions, primary sedimentation and diagenesis of sediments in such environments (Diaz and Rosenberg, 1995; Dräger et al., 2017; Nürnberg, 2004; O'Reilly et al., 2015; Shatkay et al., 1993; Tyson and Pearson, 1991; Wetzel, 2001). The development of hypoxic conditions in lakes, [$O_2$] < 2 mg/l (Nürnberg, 2004; Tyson and Pearson, 1991; Vaquer-Sunyer and Duarte, 2008), during the last ~200 years is a threat for ecological and sedimentary systems. It is driven by both natural and anthropogenic pressure (Diaz and Rosenberg, 2008; Jenny et al., 2016b), such as climate warming (Jane et al., 2021; Jankowski et al., 2006; Meire et al., 2013; Njiru et al., 2012), enhanced water column stratification and decreased lake circulation (Jankowski et al., 2006; Straile et al., 2003), nutrient input increasing primary productivity and decomposition of organic matter (Dräger et al., 2017; Kienel et al., 2013), and water-sediment interactions (Steinsberger et al., 2017). Hypoxic conditions in lakes support the burial and preservation of organic matter (OM) in lacustrine records over $10^2$-$10^3$ yrs time scales (Arthur and Dean, 1998; Dräger et al., 2017). The burial of OM in lacustrine sediments contributes to the global carbon fixation, and it is estimated to reach half the carbon fixation in the oceans (Dean and Gorham, 1998; Kastowski et al., 2011; Mendonça et al., 2017; Mulholland and Elwood, 1982; Tranvik et al., 2009), due to the high OM productivity in lakes (Tyson and Pearson, 1991), and the high burial efficiency under hypoxic conditions (Sobek et al., 2009). To improve evaluation of organic carbon fixation in lake sediments, and how the efficiency of organic carbon fixation in lakes changes between oxygenated and hypoxic intervals, an accurate and high-resolution quantification of the rate and intensity of hypoxia spread in lakes and a good evaluation of OM productivity, accumulation and preservation in lake sediments are required.

Over the years, attempts to reconstruct past oxygen level (paleoredox conditions) in lakes used diverse sedimentological, geochemical and biological proxies (Anderson and Dean, 1988; Buatois et al., 2020; Dräger et al., 2019; Dräger et al., 2017; Friedrich et al., 2014; Jenny et al., 2013; Makri et al., 2021; Ojala et al., 2000; Sorrel et al., 2021; Teranes and Bernasconi, 2005; Ursenbacher et al., 2020). Laminae preservation along lacustrine successions indicates the absence of bioturbating organisms, while non-laminated intervals reflect DO level sufficient for the existence of bioturbation organisms in cases where the homogenous texture do not originate from mass transport deposits (Diaz and Rosenberg, 1995; Friedrich et al., 2014; Jenny et al., 2013; Kelts and Hsü, 1978; Kienel et al., 2013; Ojala et al., 2000; Schaffner et al., 1992; Tylmann et al., 2012; Tyson and Pearson, 1991; Zolitschka et al., 2015). Low oxygen conditions in the sediment-water interface during laminated sediment intervals are further supported by high bulk sediment TOC concentrations within these sediments due to the reduced organic matter (OM) decomposition (Arthur and Dean, 1998; Diaz and Rosenberg, 1995; Dräger et al., 2017). As a result, a more negative $\delta^{13}C_{org}$ composition is measured on laminated intervals with higher TOC content because OM degradation selectively degrade organic compounds with a more negative $\delta^{13}C$ composition, thus the $\delta^{13}C_{org}$ value of the remaining OM fraction in non-laminated intervals becomes less negative (Benner et al., 1987; Dräger et al., 2017; Lehmann et al., 2002; Mollenhauer and Eglinton, 2007; Spiker and Hatcher, 1987). Therefore, $\delta^{13}C_{org}$ analyses allow to determine whether high TOC concentrations increased production or preservation. An additional method for reconstructing oxygen level in lakes is by analyzing the abundance of redox-sensitive elements in the sediments. X-ray fluorescence (XRF) scans of sediment cores (Evans et al., 2019; Makri et al., 2021; Sanchini et al., 2020; Sorrel et al., 2021; Zander et al., 2021). The element data allow a continuous and high-resolution reconstruction of paleoredox conditions, rather than the binary laminated non-laminated perspective. Specifically, the ratio between Fe and Mn ratio is often used for paleoredox reconstructions because of the differential re-mobilization of Fe and Mn under redox conditions (He et al., 2023; Loizeau et al., 2001; Makri et al., 2021; Naeher et al., 2013; Żarczyński et al., 2019), although the Fe/Mn ratio remains a proxy of relative changes of the sediment redox state. The abundance of oxygen-sensitive bioturbating organisms in lakes was used for DO level reconstruction (e.g., Ursenbacher et al., 2020). In temperate regions, chironomid species during their larval stage serve as one of the main bioturbating organisms in lakes. Remains of these larvae preserve well in the sediments and their analysis therefore potentially allows reconstructions of past DO concentrations as well and to link the spread of hypoxic conditions to laminated intervals (Brodersen et al., 2004; Davies, 1976; Heinis and Davids, 1993; Ursenbacher et al., 2020). However, the abovementioned methods commonly were applied on a single sediment core from the deepest part of the lake basin, so that the rate of the spatiotemporal spread of hypoxia remained unknown. In this study, we apply a multi-proxy approach on multiple sediment cores from the entire lake basin at different water depths.

Annually laminated or varved lake sediments are unique high-resolution archives to reconstruct the intensity of hypoxia and evaluate the rate in which hypoxic conditions spread (Dräger et al., 2019). Reconstructing the rate and extent of hypoxia in lakes provide a valuable basis for sustainable development, biological conservation and evaluating anthropogenic pressure on the environment (Jenny et al., 2016b; Njiru et al., 2012). This study addresses open questions on the spatiotemporal evolution of hypoxia within a lake and critical conditions for initiating preservation of varved sediments. The main objective of this study is to reconstruct the spatiotemporal

spread of hypoxia in lake Tiefer See during the last two centuries in detail, and to decipher the response of the sedimentary system to depleted DO levels. Moreover, we wish to identify thresholds for hypoxia spread in the lake using present-day monitoring data. Finally, we want to test the common approach of using varve preservation as proxy for a hypoxic lake regime. Our approach combines micro-facial and geochemical analyses

of 17 gravity sediment cores from different water depths and locations within the lake basin with lake water monitoring data. The results are supported by paleoecological analyses (subfossil chironomids) from selected samples of two of the cores to support the interpretation of changing DO levels at these sites.

## 2    Site description

Lake Tiefer See (TSK; 53º 35.50'N, 12º31.80'E; 62 m a.s.l) is located in the northeastern German lowlands

(**Figure 1Figure 1**A) and part of the Klocksin lake chain, and was formed within a subglacial channel system at the end of the last glaciation. It has an elongated axis oriented north-south with steep slopes on the east-west directions (**Figure 1Figure 1**B) and has a surface area of 0.75 km$^2$. The modern lake has a maximum depth of 62 m and is an ideal site to link environmental conditions to the preservation of laminated sediments (Roeser et al., 2021). The lake has relatively wide and shallow margins (**Figure 1Figure 1**C) and the deepest part of the lake

(31-62 m) includes only small part of the lake's area (~16 %) and water volume (~10 %). Only negligible inflow enters the lake directly from lake Flacher See and the small catchment area (~5.2 km$^2$) is dominated by glacial till of the terminal moraine. This area is presently used as arable land, whereas the lake shorelines are covered by a narrow band of large alder, ash, and oak trees (Kienel et al., 2013).

The TSK is a mesotrophic-monomictic lake with a stratified water column from March to October, while the

105 water column is well mixed from November to February (Roeser et al., 2021). This limnologic mode has a significant impact on lake's circulation, oxygenation of the water column and endogenic mineral formation. Present-day annually laminated sedimentation comprises an early spring diatom sub-layer, a late spring-early summer endogenic calcite sub-layer (Kienel et al., 2017; Roeser et al., 2021), and an autumn-winter mixed layer of organic matter and littoral calcite sub-layer (Roeser et al., 2021). The three layers closely follow the annual

stratification and oxygenation of the lake with thermal stratification strengthening during spring and summer and a well-mixed water column during autumn and winter. Allogenic sedimentation, such as dust influx and sediment transport by surface runoff is negligible. In the recent two centuries the lake surroundings experienced an increased anthropogenic pressure that led to an increased nutrient flux into the lake from anthropogenic fertilizers and manure (Kienel et al., 2013). As a result, the lake experienced eutrophication with an increased

OM flux and seasonal diatom blooms (Kienel et al., 2017). Following the increasing OM flux to the lakefloor, the consumption of oxygen at the lakefloor increased substantially leading to prevailing hypoxic conditions.

The TSK Holocene sediment record is characterized by frequent alternations between varved and homogeneous non-varved intervals (Dräger et al., 2019; Dräger et al., 2017). Varved intervals are associated with high TOC content of ~15% and more negative $\delta^{13}C_{org}$ (-32‰), while sediments of non-varved intervals characterized by a

120 lower TOC content of ~5% and a less negative $\delta^{13}C_{org}$ (-28‰) (Dräger et al., 2019). The preservation of

varved interval is attributed to the prevailing hypoxic conditions (Dräger et al., 2017), and indicate that the TSK is sensitive to various processes that regulate oxygen levels along the water column.

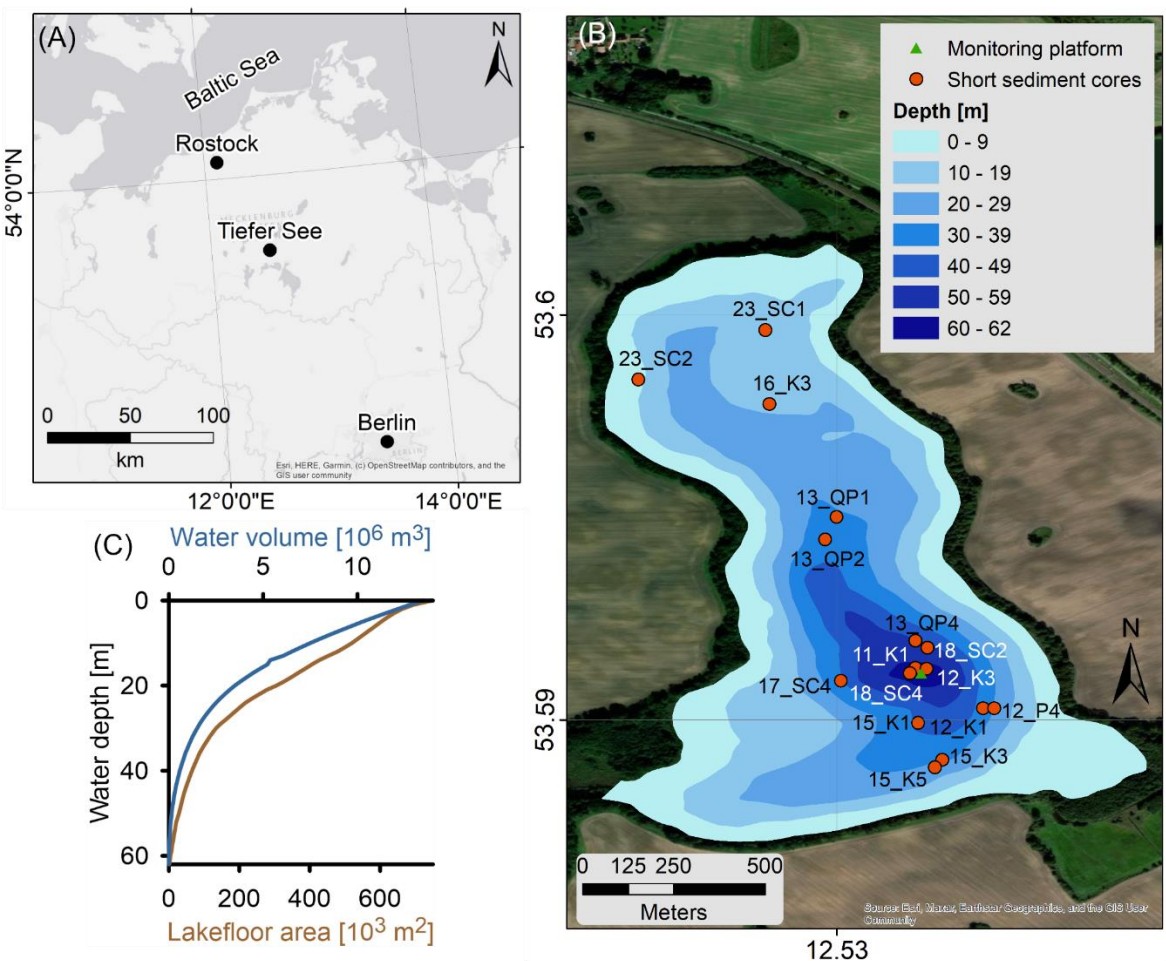

**Figure 1: Location map of the TSK. (A) The lake is located at the Southern Baltic region, occupying a subglacial channel system. (B) TSK bathymetry, short cores and monitoring system locations. (C) The hypsometric curve of the lake. Imagery photo source: ESRI, Earthstar Geographics, and the GIS user community.**

## 3    Methods and materials

In this study we apply a novel combination of three independent high-resolution proxies for hypoxia; (i) varve preservation and counting based on microfacies analyses that provides a robust time constrain, and (ii) high resolution geochemical depth profiles that allow to quantify the spread of hypoxia in the lake. (iii) Faunal remains as indicators for potential burrowing organisms. This combination is applied for sediment cores from different water depths and locations within the lake basin (**Figure 1**). For selected cores and intervals, we also analyzed the remains of chironmid larvae in the sediments, one of the main bioturbating organism groups in TSK that is expected to change in abundance and species composition with changing DO. In order to quantify present-day seasonal variations in hypoxia, we use lake water monitoring data. All analyses were conducted at the GFZ, Potsdam, Germany except of the chironmid analysis that was carried at the geoecology group at the University of Basel.

## 3.1 Sediment cores

In total 17 short cores were taken with a 90 – mm UWITEC gravity corer (Dräger et al., 2019) during the period of 2011-2023 at different locations and water depths, between 10-62 m (**Figure 1**), covering the recent varved interval in the lake. Each one of the cores is documented by the location, water depth [m], core length [cm] and thin section length [cm]. After recovery, the cores were left settling in a cold room for two weeks in a vertical position at a temperature of 4°C before core opening and sampling (Roeser et al., 2021).

## 3.2 Micro-facies analyses

From 16 of the cores, 46 overlapping thin sections (100 x 25 mm) were prepared following the standard procedure for soft sediments based on the freeze-drying technique and impregnation with epoxy resin (Brauer and Casanova, 2001). The thin sections were used for varve counting based on microscopic identification of seasonal layers to achieve a robust chronology of the upper varved unit. For an overview, thin sections were first scanned on a standard flatbed scanner with 1200 dpi resolution. Varve counting was carried out using two different types of microscopes, a Zeiss Axio–Zoom V16 and an Olympus DP72 with non-polarized, semi-polarized and polarized light. For each core, a composite profile of series of overlapping thin sections was established by microscopic correlation. Varve counting was carried out on these continuous thin section profiles for each gravity core.

## 3.3 Chronology

The chronology of the cores is based on three methods, (i) varve counting, (ii) core correlation by marker layers, and (iii) tephrochronology. Varve counting and core correlation were applied to acquire the chronology of the laminated interval, and tephrochronology was used to acquire an age anchor in the non-laminated sediments.

### 3.3.1 Varve counting, marker layers identification and core correlation

We adopt the varve facies model from Roeser et al. (2021) for varve counting in large scale (10 cm long) thin sections using a petrographic microscope. Varve counting along the Holocene record of TSK was proven reliable, and was validated by radiocarbon dating and tephrochronology (Brauer et al., 2019), thus we adopt this technique. Marker layers (ML) are distinct layers or sequences of laminae that can be identified in thin sections (only few are also macroscopically visible) and serve for core correlation. ML are considered as isochrons and their age is determined by varve counting. MLs were initially defined in core TSK11-K1 due to its excellent recovery and its location in the deepest part of the lake (**Figure 1**), with a single ML that was identified in core TSK18-SC4 that allows to complete the varve counting until 2018, thus covering the entire recent hypoxic period. The age uncertainty is estimated by the uncertainty of the number of varves counted in core TSK11-K1 between two consecutive ML. Due to the very good preservation of the varves the dating uncertainty is below 1%.

### 3.3.2  Tephrochronology

Tephrochronology was applied for three cores from intermediate water depths (TSK12-K1 from 35.1 m; TSK13-QP1 water depth 30.3 m; TSK15-K5 water depth 27.3 m) to identify the Askja (Iceland) eruption from CE 1875 as an independent isochrone. Previous work by Wulf et al. (2016) has identified the Askja-1875 as a crypto-tephra layer in the core TSK11-K1. From each of the cores in this study, five continuous 1-cm-interval samples were analyzed from the age-depth interval which is expected to include the Askja-1875 tephra. The extraction of these microscopic glass shards from the sediment samples used the adapted methods outlined in Blockley et al. (2005). The volcanic glass shards were then identified via microscopy before being physically extracted using the method outlined in Lane et al. (2014). The major and minor elemental composition of the individual shards were measured using electron probe microanalysis (EPMA) using the JOEL JXA-8230 at GFZ Potsdam. The operating conditions were as followed: a beam size of 5-10µm with 15kV voltage and a 10 nA beam current. The count times for Fe, Ti, Mg, Mn, Cl and P were 20 s and for Si, Al, K, Ca and Na were 10 s. The machine was calibrated using the glass standards of MPI-Ding glasses ATHO-G, StHs-6-80 and GOR-132-G (Jochum et al., 2006) and Lipari obsidian (Hunt and Hill, 1996) with measurements taken on these glasses during the run to ensure precision and accuracy.

## 3.4  Geochemical sediment characterization

### 3.4.1  XRF scanning

Element maps were acquired on impregnated sediment blocks, which are the residual counter part of the thin-sections, using a Bruker M4 Tornado micro-XRF scanner. The XRF element maps allow direct comparison of the geochemical composition and thin-section observations of individual layers. Elemental XRF mapping analyses were performed in a vacuum chamber at 50 µm resolution with a Bruker M4 Tornado micro-XRF scanner. This micro-XRF scanner is equipped with a Rh X-ray source that is operated at 50 kV and 0.60 mA. The poly-capillary X-ray optics produces a high intensity irradiation spot of about 20 µm that allows fast measurement times of 50 ms.

Continuous element records were acquired every 0.2 mm directly at the split core surface using an ITRAX XRF core scanner. These non-destructive analyses were performed with an Rh X-ray source (30 kV, 60 mA) for 4 s after careful cleaning covering the core surface with XRF-transparent foil to prevent desiccation of the core during the measurements. Additional to the continuous XRF measurements, 2 to 3 selected intervals of ca 2 cm long were selected to obtain replicate measurements.

Generally, XRF core scanning measurements generally cover the elements Aluminium (Al) through to Zircon (Zr), but only the elements Si, S, K, Ca, Ti, Mn, Fe, and Sr were selected based on the relative standard deviations (<25%) calculated from 3-fold replicate measurements. Element intensity records (in cps) contain information of the relative element concentration, but are also influenced by changes of physical sediment properties and matrix absorption and enhancement effects (Tjallingii et al., 2007). However, log-ratios of element intensities are linear functions of log-ratios of concentrations, and also allow consistent statistical

analysis of compositional data (Weltje et al., 2015). Therefore, statistical analyses were performed after center-log-ratio transformation (Bertrand et al., 2024) of the selected element intensities using the Xelerate software package (Weltje et al., 2015). Element correlations were explored using a d PCA biplot and Ward's hierarchical clustering was used for geochemical characterization of sediment cores. The statistical analyses were performed on the XRF data of all the selected cores to reveal similarities and differences for all sediments within the lake system.

### 3.4.2    TOC, $\delta^{13}C_{org}$ and CaCO$_3$ determination

Five short cores (TSK11-K1, TSK12-K1, TSK13-QP1, TSK15-K5 and TSK23-SC2) were sampled for the purpose of determining TOC, $\delta^{13}C_{org}$ and CaCO$_3$ profiles at contiguous 1 cm$^3$ samples.

Total carbon (TC), total organic carbon (TOC) and $\delta^{13}C_{org}$ were determined using an elemental analyzer (FlashEA 1112) connected with a ConFloIV interface on a DELTA V Advantage IRMS (isotope ratio mass spectrometer, ThermoFischer Scientific) at the GeoForschungsZentrum Section 4.3 in Potsdam, Germany. For TC up to 2 mg sample material was loaded into tin capsules and combusted in the elemental analyzer. The calibration was performed using Urea and checked with a soil reference sample (Boden3, HEKATECH). The TOC contents and $\delta^{13}C_{org}$ values were determined on in-situ decalcified samples. Around 3 mg of sample material were weighted into Ag-capsules, dropped first with 3 % and second with 20% HCl, heated for 3 h at 75°C, and finally wrapped and measured as described above. The calibration was performed using elemental (Urea) and certified isotope standards (IAEA-CH-7) and checked with internal reference sample. The isotopic composition is given in delta notation relative to a standard: $\delta$ (‰) = [(R$_{sample}$ − R$_{standard}$)/R$_{standard}$)] x 1000 and the reproducibility for replicate analyses is 0.2 % for TOC and 0.2‰ for $\delta^{13}C_{org}$. Calcite contents were calculated by obtaining the total inorganic carbon content (TIC = TC − TOC) and multiplying by 8.33, which is the fraction molar mass of inorganic carbon from the CaCO$_3$.

### 3.5    Chironomid analysis

Abundance and taxonomic composition of chironomid remains were analyzed for selected subsamples from varved and non-varved sediment sections in two sediment cores, the shallow TSK15-K5 (27.3 m) and the deep TSK18-SC4 (61.1 m). Six sediment samples were analyzed in each core, with 2 cm³ of sediment processed for all samples except the deepest sample in core TSK-K5, where only 1 cm³ was analyzed. Three samples were taken from the top varved unit and three from the non-varved unit below. Sediments were washed through a 100-µm sieve and the sieve residue then examined under a stereomicroscope (Leica DM2500) at 20-40x magnification using a Bogorov counting tray. Chironomid remains were sorted from other remains, mounted in Euparal mounting medium and identified at 100-400x magnification under a compound microscope. The remains were identified to morphotypes according to Brooks et al. (2007). Larval remains were classified into taxa that can colonize deepwater (profundal) environments in lakes and taxa restricted to shallow water (littoral environments) based on Brooks et al. (2007), Wiederholm (1983) and Saether (1979). Concentrations were calculated by dividing chironomid counts by the analyzed sediment volume and influx values by multiplying concentrations with estimated sediment accumulation rates in cm/yr. Interpretations focus on the overall

concentrations, percentage abundances and influx values of the sum of profundal chironomids and selected profundal chironomid taxa.

### 3.6 Monitoring data

An observational setup based on a research platform at the deepest part of the lake used for high-resolution limnological-sedimentological monitoring since 2012 and provides that data used in this study (Roeser et al., 2021). Temperature and oxygen profiles of the water column were measured every 12 hr with an automatic water probe (YSI 6600 V2, Yellow Springs United States). This data is used to trace the evolution of the summer thermal stratification over time. CaCO$_3$ and TOC contents, and $\delta^{13}C_{org}$ of modern sediments were measured on bi-weekly sediment samples collected by a sediment trap at a water depth of 50 m. The sediment sampling was conducted at intervals of 15 days with an automated sequential trap (Technicap PPS 3/3; active area 0.125 m$^2$) equipped with 12 sample bottles.

## 4 Results

### 4.1 Cores stratigraphy

15 cores retrieved from a water depth >12 m contain the uppermost varved interval (**Figure 2**). Two cores retrieved from less than 12 m water depth (TSK23-SC1/SC2) do not contain a varved unit. Except the two cores without a varved unit all cores depict a similar sediment sequence consisting of three units: The basal Unit B composed of homogeneous dark brown sediments is overlain by Unit G of homogenous grayish sediments followed by the varved Unit V on top. The thickness of Unit V ranges between 4.5 cm in the shallowest core (TSK16-K3) from 16 m water depth, and 40 cm in the deepest core (TSK11-K1) from 62 m water depth. In contrast, Unit G is thicker in the shallow water cores (13-15 cm) than in the depocentral cores (2 cm).

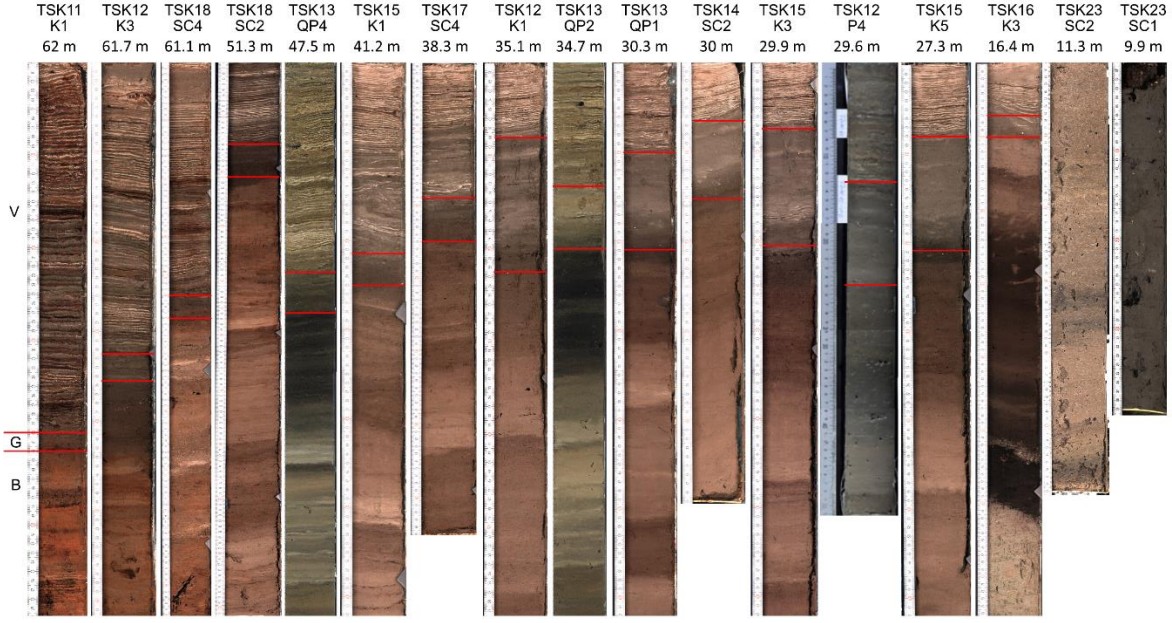

**Figure 2. Core images of all short cores used in this study arranged form deep to shallow. V – varved unit at the top of each core, G – gray unit, B – brown unit.**

### 4.2    Varve composition

The upper laminated unit is mainly composed of biogenic-calcareous varves as defined in the lake by Roeser et al. (2021) as triplet of diatom-rich, calcite-rich and resuspension sub-layers (**Figure 3**). Micro-XRF maps display the abundance of Ca and Si, as indicators for the calcite and diatom layers. They show that these elements concentrate along distinct laminae, emphasizing the seasonal deposition of calcite and diatoms. While the diatom and calcite sub-layers are very distinct in the Micro-XRF scans due to the high abundance of Ca and Si,

the mixed layer is rich in organic materials with only minor calcite crystals and diatom frustules, and thus less distinct. The biogenic-calcareous varve type is deposited and preserved both at the deep and the shallow regions of the lakefloor (**Figure 3**).

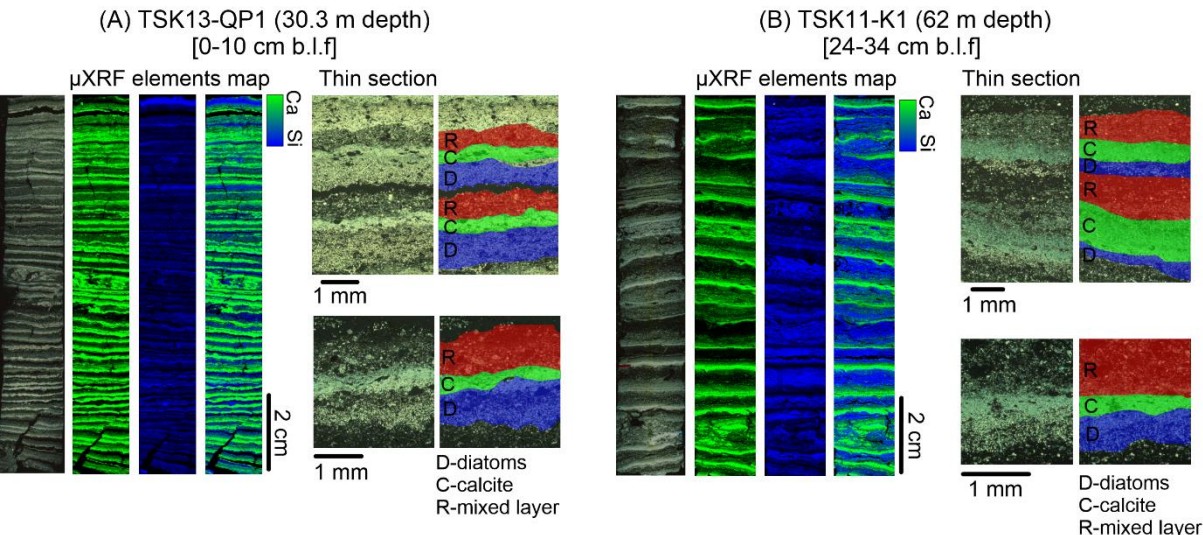

**Figure 3. Typical varves in TSK displayed by thin section images and micro-XRF mapping results of Ca and Si as**
**deposited at, (a) shallow water depth, and (B) a deeper part of the lake.**

### 4.3    Core chronology

The cores were dated by three methods, (i) varve counting, (ii) core correlation by marker layers, and (iii) tephrochronology. Varve counting and core correlation were applied on 14 cores to determine the age of the onset of varve preservation. The age of the boundary between units B and G has been obtained for four cores

(TSK11-K1 (62 m), TSK12-K1 (35.1 m), TSK13-QP1 (30.3 m) and TSK15-K5 (27.3 m)) by tephrochronology.

#### 4.3.1    Chronology of the varved sediments (Unit V)

***Varve counting and marker layers - TSK11-K1 master core.*** Core TSK11-K1 (obtained in 2011) has been selected as a master core where ten marker layers (ML1 -ML10) were defined and varve dated (**Figure 4**, Supplementary material **Table S1**). In addition, ML0 formed in 2011 was identified in core TSK18-SC4 that has been

obtained in 2018. ML0 is the anchor point of the varve chronology with varves between 2011 and 2018

counted in core TSK18-SC4 (61.1 m) and older varves in core TSK11-K1 (62 m). Varve counting in core TSK11-K1 yields a total of 93±1 varves, thus dates the onset of varve preservation to CE 1918±1 yr.

***Stratigraphic correlation.*** The onset of varve preservation was independently established in all cores and confirmed by the marker layers (**Figure 5**, supplementary material **Figure S1**). In addition to the master core, two cores, TSK12-K3 and TSK18-SC4, are also located in the deepest part of the lake, at water depths of 61-62 m. A local event layer, triggered by a thunderstorm in 2011, and occurring in core TSK18-SC4, serves as an additional marker layer (ML0) with a known age. The onset of varve preservation in the deepest part of the lake in these cores is independently obtained by microscopic varve counting at CE 1923±1 (TSK12-K3) and CE 1919±1 (TSK18-SC4), thus, within the uncertainty of the onset of varve preservation in TSK11-K1 (CE 1918±1) and an earlier published age of 1924 (Dräger et al., 2019). From the deepest part of the lake towards shallower depths, the onset

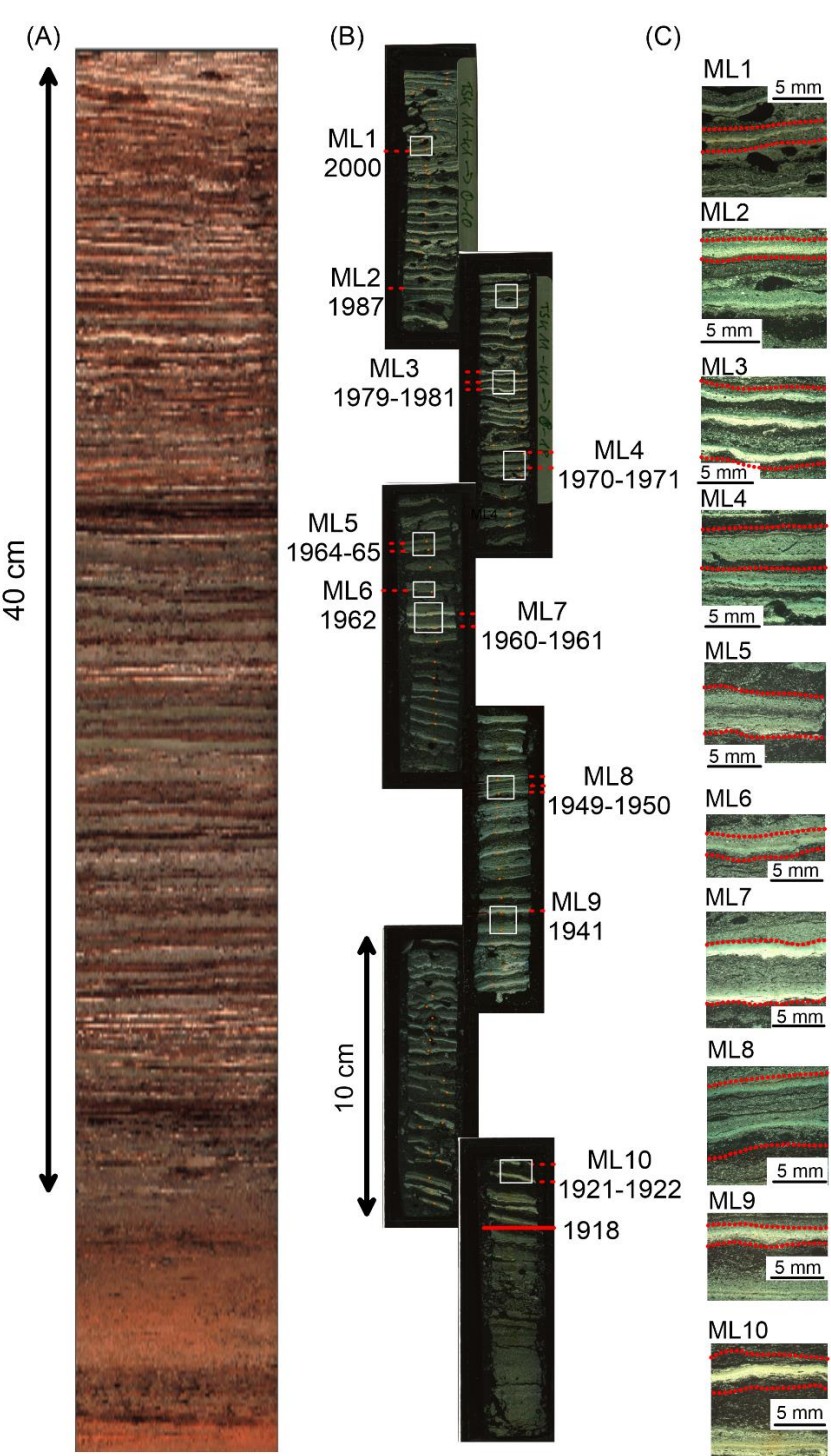

**Figure 4. TSK11-K1 master core. (A) Image of the core. (B) Thin sections of the core. (C) Marker layers (ML).**

of varve preservation becomes progressively younger (**Figure 6**, and **Table 1**). The onset of varve preservation at
TSK18-SC2 (51.3 m depth) was dated to CE 1925±1, at TSK15-K1 (41.2 m depth) to CE 1943±1, at TSK14-SC2 (30 m depth) to CE 1973±1 and at the shallowest core site, TSK16-K3 (16.4 m depth) the onset of varve preservation was dated to CE 1997±1. No varves were identified in the top sediments of the shallowest two cores, TSK23-SC1 and TSK23-SC2, from water depths of 10-12 m.

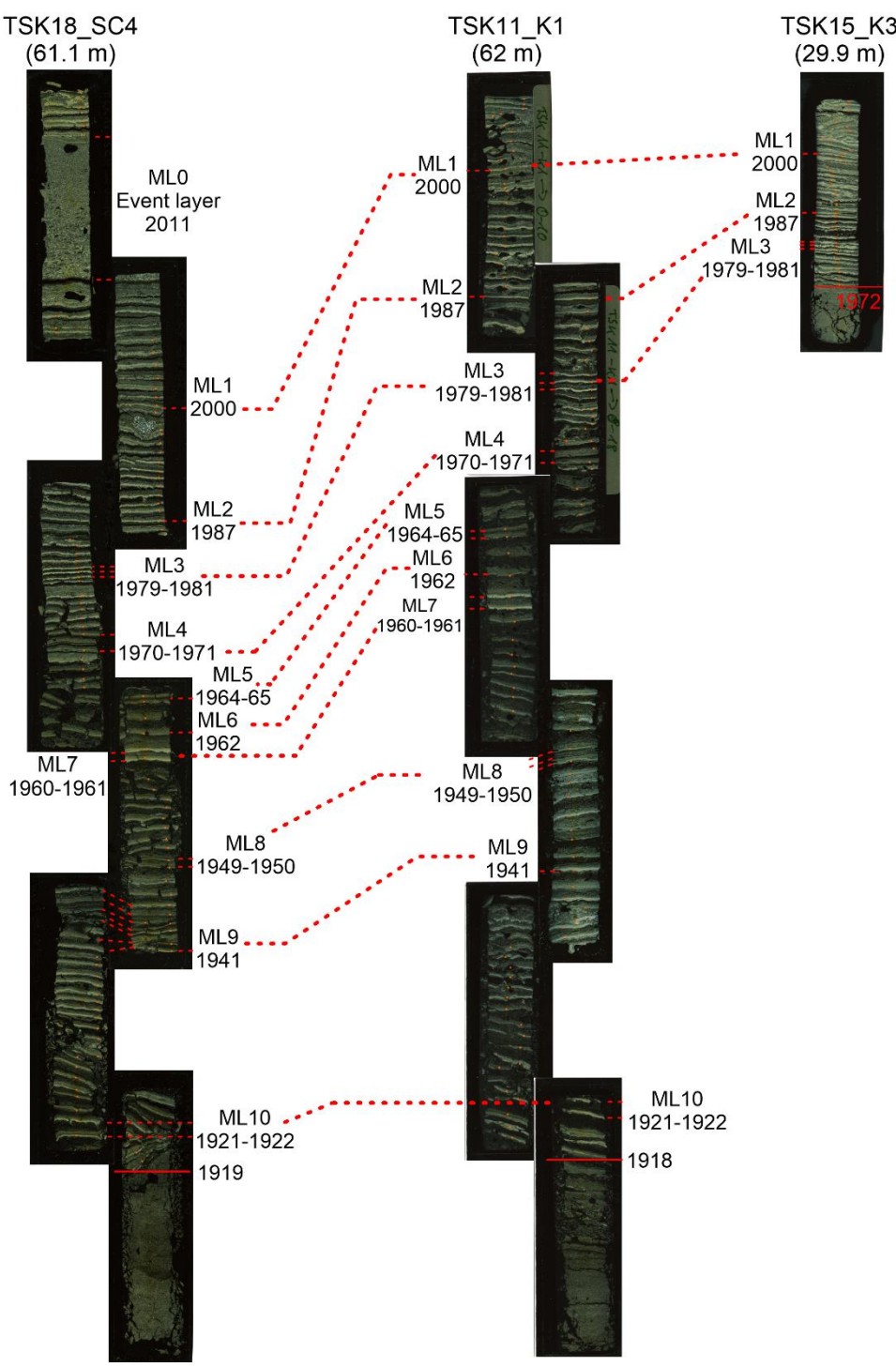

**Figure 5. Chrono-stratigraphic correlations between cores TSK18-SC4 and TSK15-K3 and the master core – TSK11-K1.**

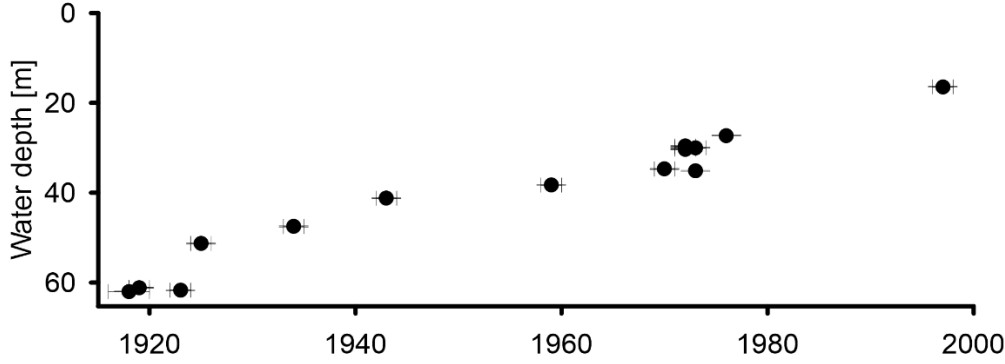

**Figure 6. Spatiotemporal spread of hypoxia in Lake TSK as shown by the onset of varve preservation and recovery depth.**

**Table 1. List of all cores that were used in the study and the age of the base of the recent varved interval.**

| | Water depth | Location | Length of varved interval [cm] | Varve onset [yr CE] |
|---|---|---|---|---|
| TSK11-K1 | 62 | N 53°35'35.60" E 12°31'48.00" | 40 | 1918±1 |
| TSK12-K3 | 61.7 | N 53°35'36.00" E 12°31'49.00" | 32.5 | 1923±1 |
| TSK18-SC4 | 61.1 | N 53°35'569" E 12°31'819" | 36 | 1919±1 |
| TSK18-SC2 | 51.3 | N 53°35'623" E 12°31'818" | 23 | 1925±1 |
| TSK13-QP4 | 47.5 | N 53°35'38.00" E 12°31'48.00" | 24 | 1934±1 |
| TSK15-K1 | 41.2 | N 53°35'512" E 12°31'804" | 21 | 1943±1 |
| TSK17-SC4 | 38.3 | N 53°35'574" E 12°31'690" | 19 | 1959±1 |
| TSK12-K1 | 35.1 | N 53°35'32.00" E 12°31'54.00" | 7.5 | 1973 |
| TSK13-QP2 | 34.7 | N 53°35'47.00" E 12°31'40.00" | 10 | 1972±1 |
| TSK13-QP1 | 30.3 | N 53°35'49.00" E 12°31'41.00" | 10 | 1970±1 |
| TSK14-SC2 | 30 | | 7 | 1973±1 |
| TSK15-K3 | 29.9 | N 53°35'457" E 12°31'840" | 8 | 1972±1 |
| TSK12-P4 | 29.6 | N 53°35'32.00" E 12°31'55.00" | 12 | 1972±1 |
| TSK15-K5 | 27.3 | N 53°35'446" E 12°31'829" | 8.5 | 1976±1 |
| TSK16-K3 | 16.4 | N 53°35'984" E 12°31'584" | 3.5 | 1997±1 |
| TSK23-SC1 | 9.9 | N 53°36'5.6154" E 12°31'34.68" | 0 | - |
| TSK23-SC2 | 11.3 | N 53°36'1.224" E 12°31'23.412" | 0 | - |

### 4.3.2  Chronology of non-varved sediments (Units B and G)

The chronology of the non-varved sediments (Units B and G) was established by tephrochronology in four cores (TSK12-K1 (35.3 m), TSK13-QP1 (30.3 m) and TSK15-K5 (27.3 m); additional age was taken for core TSK11-K1 (62 m) from Wulf et al. (2016). In each core, a peak in shard concentration was recognized and determined as isochron. Geochemical compositions show that these volcanic shards originated from the Askja-1875 CE eruption (Supplementary material Figure S2 and Tables S2 and S3). The volcanic glass shards were found in cores TSK13-QP1 (30.3 m) and TSK15-K5 (27.3 m) at the boundary between Units B and G, thus the age of this boundary is 1875. In core TSK12-K1 (35.3 m) the shards were found 3 cm below this boundary and in core TSK11-K1 (62 m) they were found 6 cm below Unit B and Unit G boundary, thus the age was estimated according to the average deposition rate of the sediments between the base of the varved unit and the identified tephra. As the crypto-tephra samples were undertaken at 1 cm that represents a certain time frame depending on the deposition rate at each core, a chronological error was added to the analysis.

### 4.4  Geochemical characterization

### 4.4.1  XRF cluster analysis

Statistical analyses were performed on the elements Si, S, K, Ca, Ti, Mn, Fe, and Sr using the Xelerate software (Weltje et al., 2015), which performs center-log-ratio transformation of the element intensity records prior to the statistical operations. Element correlations were explored using PCA biplot and geochemical characterization of the sediments was performed using Ward's hierarchical clustering for all 13 sediment cores together (**Figure 7**).

The first principal component (PC1) explains 48.3 % of the variance and reveals positive loadings for the element Ca and negative loadings for K, Ti and Fe, which reflects the variation of calcite and detrital sediments, respectively. The second principal component (PC2) explains 41.7 % of the variance with positive loadings for Mn and Fe, and negative loadings for K, Ti, and Ca, which reflects the relative variation of redox sensitive elements. Less pronounced are the loading of the element Si, which is probably related to the occurrence this element both in diatom frustules and siliciclastic detritus. Based on these element correlations we selected the element ratios of Ca/Ti, Si/Ti and Fe/Mn as proxies for relative variations of carbonate and detrital sediments, diatom and detrital sediments, and intensity of the redox-processes. These proxies are indicted as log-ratios (Figure 9) including 95% confidence intervals as calculated from the 3-fold replicate measurements.

Geochemical characterization using Ward's hierarchical clustering was performed to obtain the minimum number of statistical clusters that match the observed sedimentological units defined (V, B, and G), which are five groups in this case. These statical clusters are calculated based on the Euclidian distance of the data points to one of the 5 evenly distributed centroids in the data set without stratigraphical constrains.

Element correlations show a clear distinction between detrital sediments (Ti, K, Fe, Si), endogenic components (Ca, Si), and redox sensitive elements (Fe, Mn), whereas cluster analysis shows the difference between homogenous oxidized sediments (1-3) and varved reduced sediments (4-5). Cluster 2 represent the base of the shallow cores that composed of oxidized sediments containing higher detrital fraction; thus, it lacks Mn and is

enriched in Ti, K and Fe of detritic origin. Cluster 3 represent the base of the shallow cores that composed of oxidized sediments with higher amounts of detrital components. Cluster 4 represents the top of the section in the deep-water cores that are composed of Ca-rich reduced sediments. Cluster 5 represents the top of the section in the shallow-water cores that are composed of Ca-rich reduced sediments. Cluster 1 is glacial till (non-lacustrine) that appears at the base of one core (TSK16-K3). At the site of this core the Holocene lacustrine section is relatively thin, thus a short sediment core includes the late glacial till ((Theuerkauf et al., 2022); See supplementary material Figure S3).

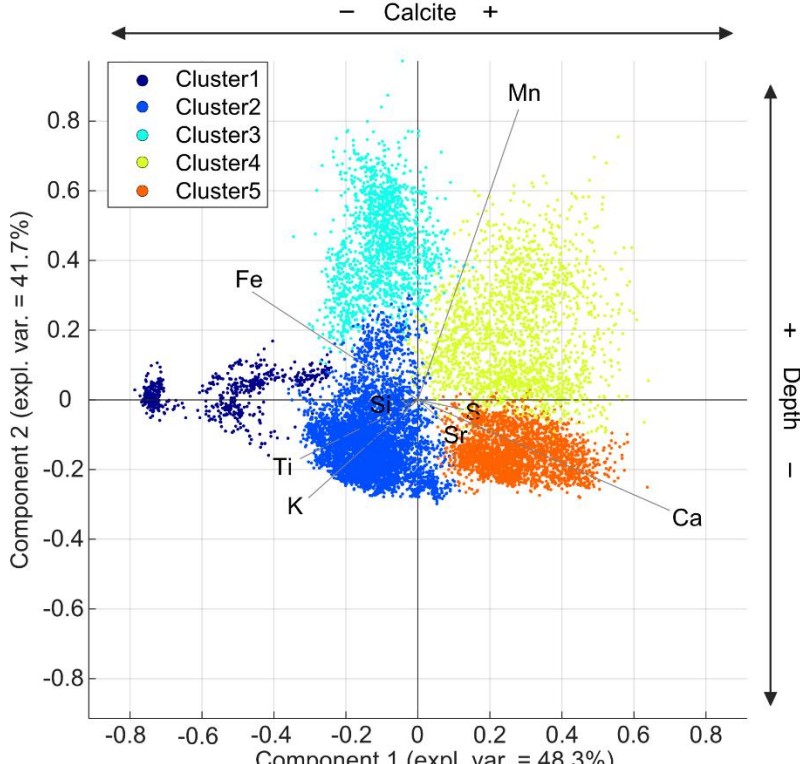

**Figure 7. PCA biplot of PC1 and PC2 showing all XRF data points and correlations of the elements measured. Geochemical characterization of the sediments by the 5 statistical clusters is indicated by the distribution of the clusters in the biplot reflecting the different sediment compositions.**

Down-core cluster representations of the data points can be used for geochemical characterization of the different sediment types, and reveal a clear change from oxic sediments of Unit B to hypoxic sediments of Units G and V (**Figure 8**). Besides, the clustering results divide between cores from shallow locations of 11-41 m water depth and cores from the depocenter, from at 51-62 m water depth. For the deeper cores, Unit B is classified as cluster 3 (Fe-Mn rich oxidized detrital sediments) while the upper Units G and V are classified as cluster 4 (Ca rich reduced sediments). For more shallow cores, unit B is classified as cluster 2 (Fe-Ti rich oxidized detrital sediments) while the upper Units G and V are classified as cluster 5 (Ca rich reduced sediments). However, for all cores the change from Unit B to the upper Units G and V is mainly related with the increasing dominance of Ca (**Figure 8**). In all of the cores, except of TSK23-SC2 (11.3 m), the transition from unit B to unit G is correlated with a substantial change in the geochemistry of the sediments marked by a

transition between cluster 2 (oxidized sediments with higher detrital components) and cluster 5 (calcite rich reduced sediments) in the shallow cores or between cluster 3 (oxidized sediments with higher detrital components) to cluster 4 (calcite rich reduced sediments) (deep cores).

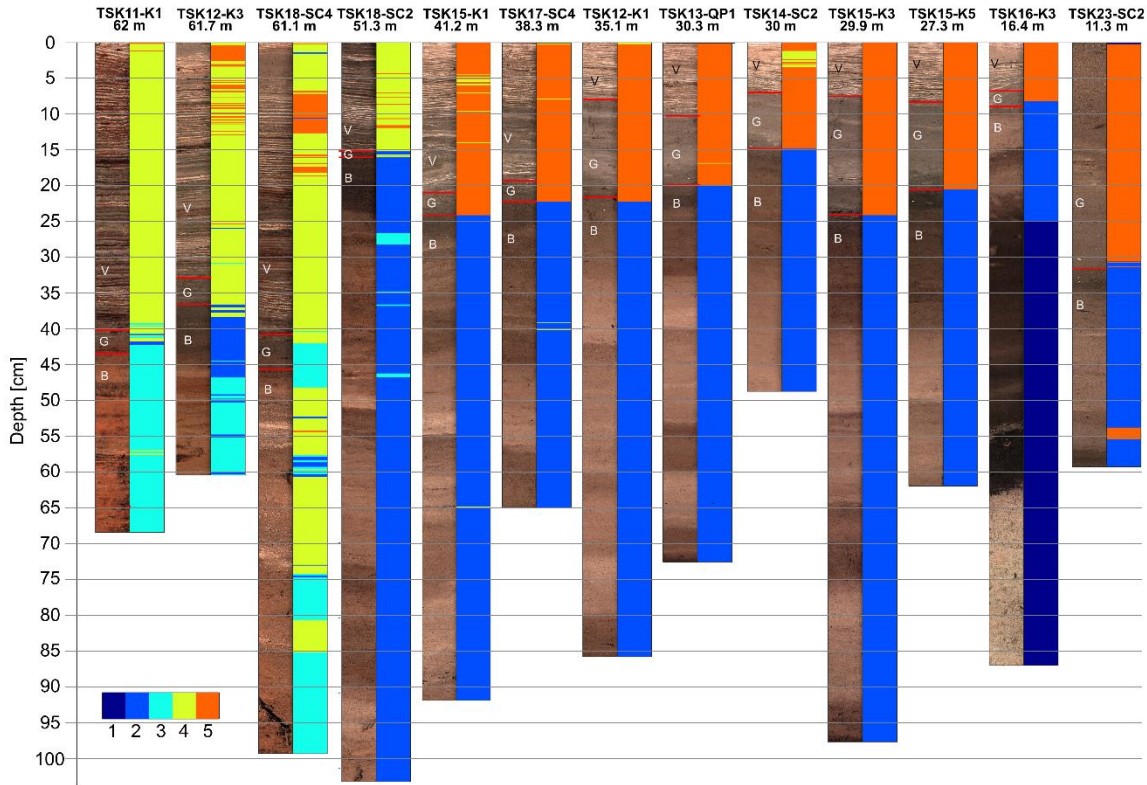


**Figure 8. XRF stratigraphy of the cores. The colors of the clusters represent the geochemical classification of the sediments based on the cluster analysis presented in Figure 7. B - brown unit. G – gray unit. V – varved unit.**

### 4.4.2    Geochemical sediment core profiles

Three cores, from different water depths (TSK11-K1 from 62 m, TSK15-K5 from 27.3 m and TSK23-SC2 from
11.3 m) were selected for detailed TOC, $\delta^{13}C_{org}$ and CaCO$_3$ analyses, and are presented here with XRF Ca/Ti, Si/Ti and Fe/Mn ratios (**Figure 9**). In core **TSK11-K1** (62 m) (**Figure 9A**), CaCO$_3$ content along Unit B ranges between 17-41% with an average of 28%, and it increases along the thin Unit G and the varved unit to 25-62% with an average of 44%. The TOC content moderately increases from ~3% at the bottom of the core to 11% at its top with no abrupt changes at the transition from Unit B to Unit G and at the transition to the varved Unit V.
$\delta^{13}C_{org}$ values decrease from ~-28‰ at the bottom of the core to ~3-1‰ at the top of the core. While the change in TOC content upcore is moderate, a stepwise increase and a stepwise decrease is observed in CaCO$_3$ content and the $\delta^{13}C_{org}$ values respectively. Both roughly coincide with the transition from Unit B to Units G and V. The abrupt changes in CaCO$_3$ content and the $\delta^{13}C_{org}$ composition are coincided with changes in the XRF elemental ratios. Ca/Ti and Si/Ti ratios are both low along Unit B and they are substantially higher in the
varved Unit. The Fe/Mn ratio displays a stepwise decrease at the transition from Unit B to Units G and V.

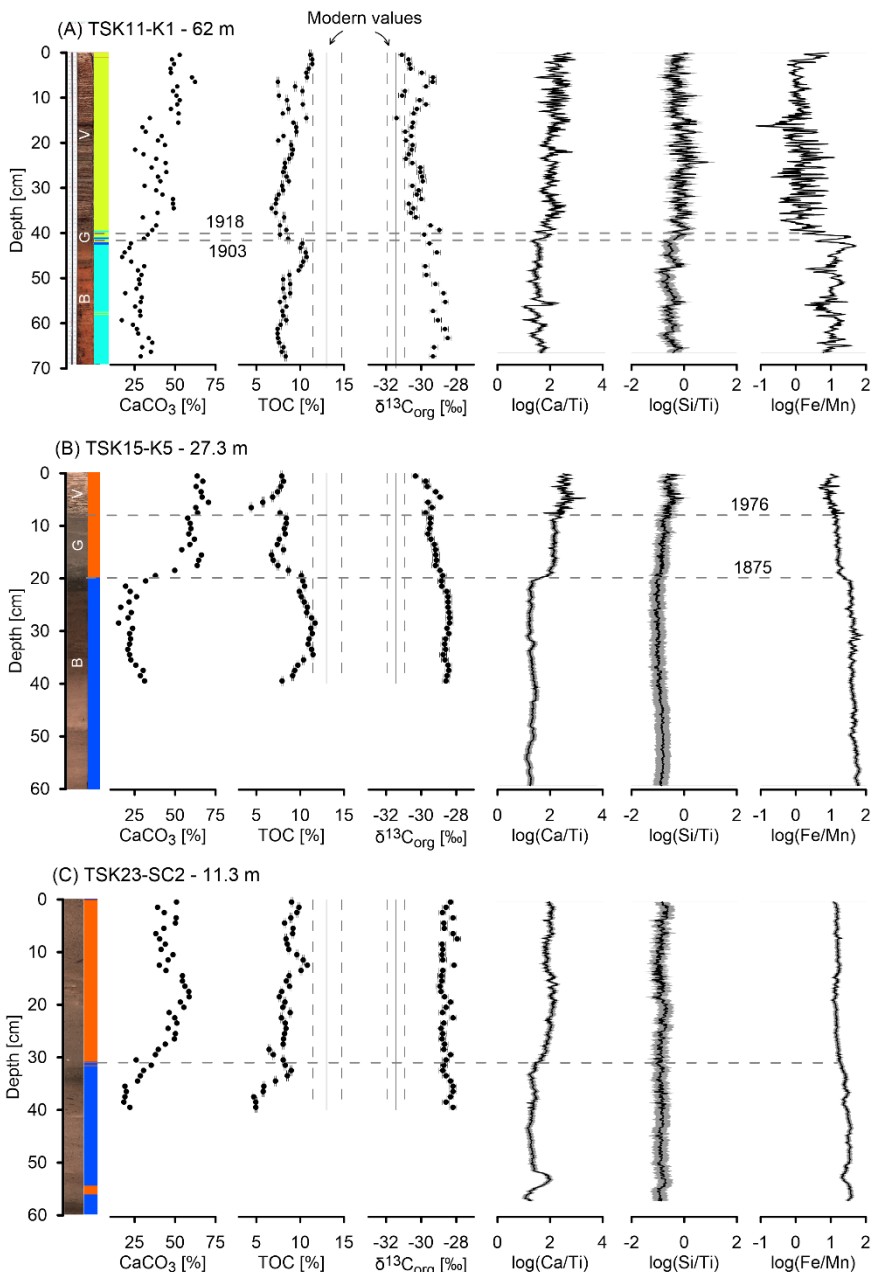

**Figure 9.** Core image, XRF stratigraphy, CaCO3, TOC, $\delta^{13}C_{org}$, **Ca/Ti, Si/Ti, Fe/Mn, profiles of three cores from different water depths. (A) TSK11-K1 at 62 m depth, (B) TSK15-K5 at 27.3 m depth, and (C) TSK23-SC2 from 11.3 m depth. All profiles use the same scale on the x axis to allow for comparison between different cores.**

In core **TSK15-K5** (27.3 m) (**Figure 9B**), CaCO$_3$ content ranges from ~20% along Unit B to ~60% in the Units G and V. This abrupt increase appears at the transition from the cluster 2 to the cluster 5. The TOC content decreases from ~10% at Unit B to ~6% along Units G and V. $\delta^{13}C_{org}$ values decrease from ~-28‰ at the bottom of the core to ~-30‰ at the top of the core. Both display a moderate change upcore in contrast to the abrupt change in CaCO$_3$ content. Ca/Ti and Si/Ti ratios are both low along Unit B and they are substantially

higher along Units G and V. The Ca/Ti ratio displays a stepwise increase, which is correlated with the transition

between units, while the Si/Ti ratio is moderately increasing upward from that point. The Fe/Mn ratio decreases sharply from the transition between Unit B to Units G and V upcore.

In core **TSK23-SC2** (11.3 m) (**Figure 9C**), the changes in all proxies are less distinct. CaCO$_3$ content ranges from ~20% at the base of the core to ~50% at its upper part. This increase is roughly correlated with the transition between cluster 2 and cluster 5. The TOC content increases from ~5% in cluster 2 to ~10% along cluster 5. $\delta^{13}C_{org}$ values remain constant along the core within a narrow range of -28‰ to -28.8‰, and do not show a depletion trend like the other two cores. The Ca/Ti ratio is low at the base and it is moderately increasing from the transition between the cluster 2 to cluster 5. The Si/Ti ratio do not show a clear trend upcore. The Fe/Mn ratio is constant along the core with little variance. In contrast to the other two deeper cores, the transition between the clusters (and sediment units) is not associated with a substantial change in the Fe/Mn ratio.

### 4.4.3    Chironomid abundance

Remains of chironomid larvae were present at low abundances in the analyzed cores, with the exception of the lowest sample in the intermediate water core (TSK15-K5, 27.3 m) (**Figure 10**; full dataset in supplementary material Table S4). The assemblages were characterized by a high percentage of littoral (shallow-water) taxa remains, whereas profundal (deep-water) chironomids were only present at relatively low percentages. At ~27 m water depth (TSK15-K5), the abundance, concentration and influx of remains of chironomid larvae are gradually reduced from Unit B to Unit G, and drop down to almost zero in Unit V (**Figure 10A**). In Unit B, high abundance of chironomids from profundal environments is observed, including high abundances of *Micropsectra radialis-type*, while in Unit G lower abundance of chironomids and M. radialis-type are observed and in Unit V only negligible amounts of chironomids are observed. In the deep-water core (TSK18-SC4, 61.1 m), the overall abundance of chironomids and the abundance of profundal chironomids is very low throughout the core (**Figure 10B**), in line with low oxygen concentration and low amounts of bioturbation. However, among the most abundant deepwater chironomid taxa a shift is apparent at ca. 37-30 cm depth where *M. radialis*-type and *Tanytarsus lugens/mendax*-type decline and Chironomus increases in percentages.

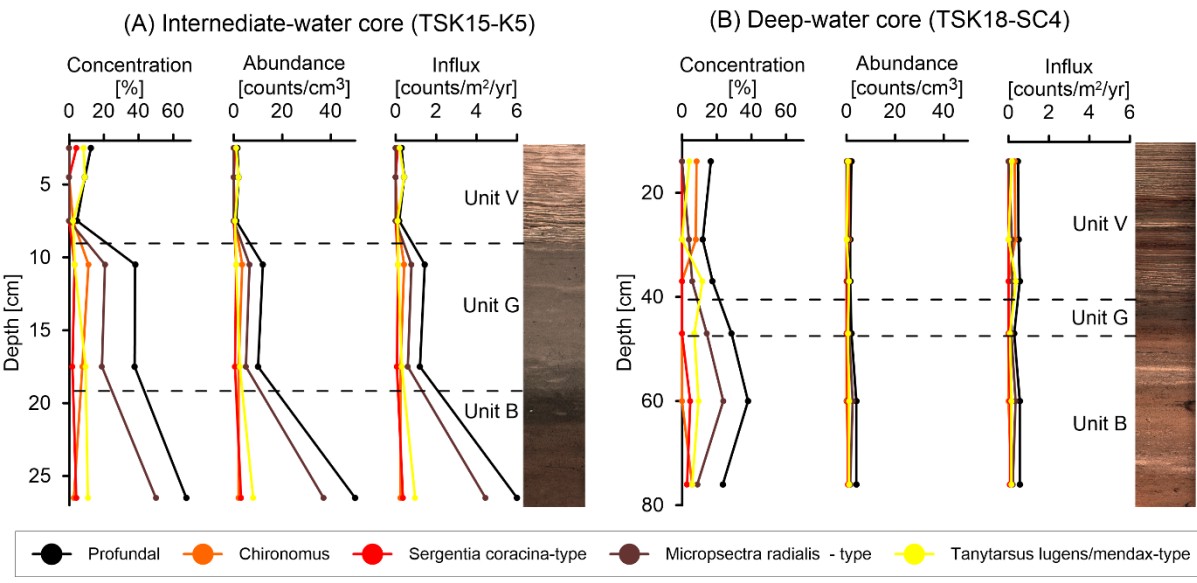

**Figure 10. Abundances of chironomids, profundal chironomids, and selected profundal chironomid groups in cores TSK15-K5 (A) and in core TSK18-SC4 (B).**

### 4.5    Limnological monitoring

The present-day DO dynamics are shown in lake monitoring data between April 2019 to February 2021 (**Figure 11B**) The dynamics of DO along the water column at the deepest part of the lake show clear seasonal fluctuations, in which DO at the upper ~12 m ranges from ~8 mg/l during the early spring to ~14 mg/l during the winter, while near the lakefloor, DO ranges from ~1 mg/l in early winter to ~11 mg/l during the spring. The depletion in DO at the hypolimnion first occurs at the water-sediment interface and prograde upward during the summer-autumn to a water depth of ~12 m (**Figure 11A**), which represents the oxycline. Thermal stratification prevails from April to November with an average thermocline depth of ~10 m (**Figure 11B**). The epilimnion temperature ranges from ~3 °C in winter to ~22 °C in summer, while hypolimnion temperature ranges from 3 °C in winter to 5 °C in summer. An interesting observation is the time shift between changes in the vertical oxygen distribution and the temperature dynamics. While the maximal temperature difference between the epi- and hypolimnion is achieved in August (**Figure 11B**), the maximal DO difference between the layers appears in November (**Figure 11B**).

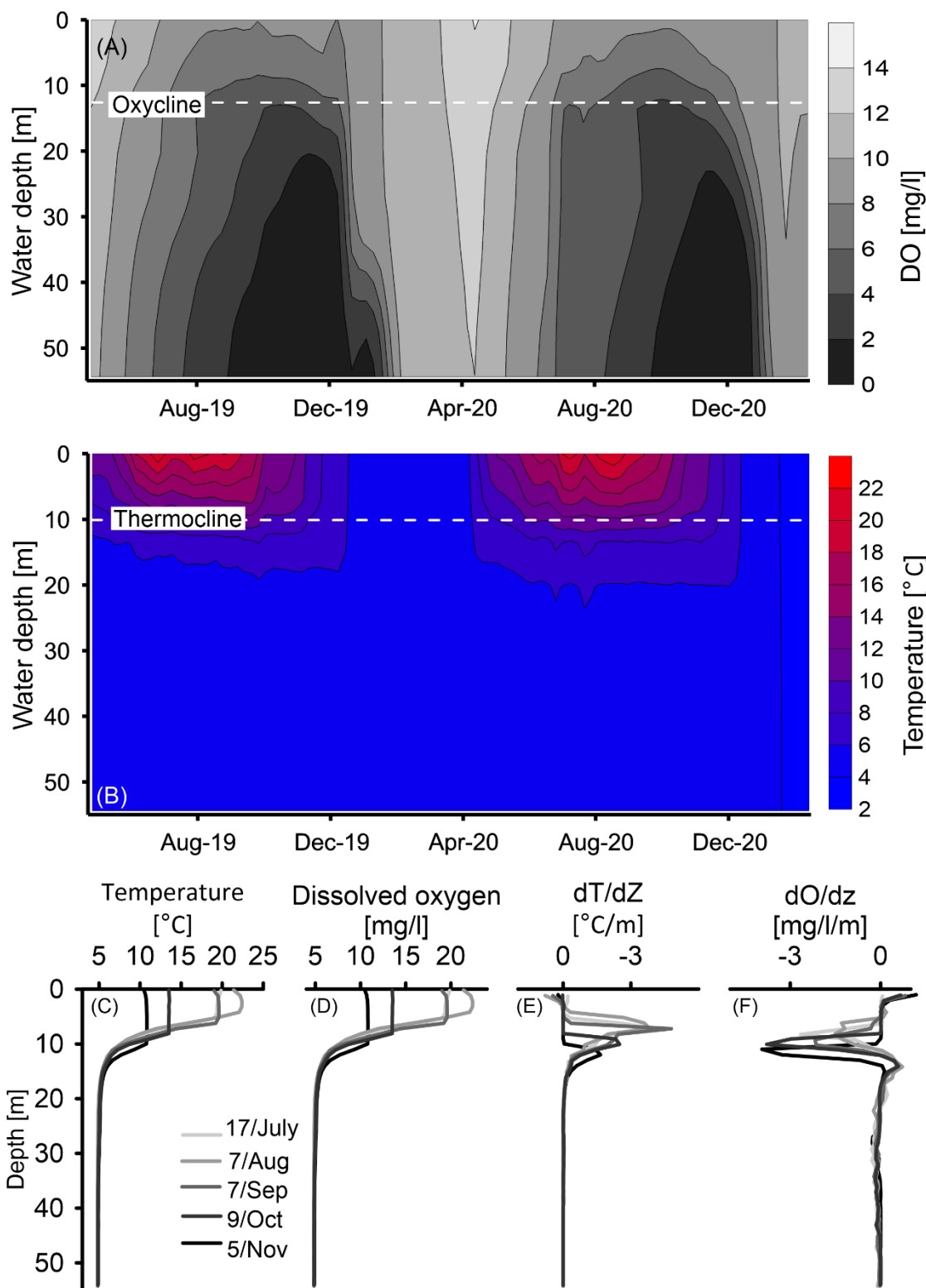

**Figure 11. Physical properties of the water column. Spatiotemporal characterization of dissolved oxygen (DO) (A) and temperature (B) in the lake. The diminishing of the temperature (C) and DO (D) stratification along the water column. The maximal depth gradient of temperature (E) and DO (F) along the water column show the evolution of the thermocline and oxycline depth.**


## 5 Discussion

### 5.1 Present-day oxygen dynamics

The seasonal water column stratification and the associated lake circulation play a main role in the bottom water
oxygenation in TSK. Present-day seasonal dynamics of DO concentrations in the lake show that hypoxia is
extending upward in the water column during summer-autumn and is limited by the thermocline around 10 m
depth (**Figure 11B**). The vertical variations of DO along the water column at the deepest part of the lake allow to
quantify the required intensity, duration and vertical development of hypoxia, which is required for varve
preservation. The general definition for hypoxia is $[O_2] < 2\frac{mg}{l}$ (Diaz and Rosenberg, 1995; Jenny et al., 2014;
Tyson and Pearson, 1991), although for freshwater lakes value of $[O_2] < 5\frac{mg}{l}$ have been used to define hypoxic
conditions (Njiru et al., 2012). By adopting these numbers, we can define the hypoxic threshold for varve
preservation at TSK below 12 m water depth by $[O_2] < 5\frac{mg}{l}$ for five months (August-December) and $[O_2] < 2$
$\frac{mg}{l}$ for two months (November-December; **Figure 11A**). Our results show that in spite of the winter oxygenation
of the water column, varves can be preserved in monomictic lakes, as long as the duration and magnitude of the
seasonal hypoxia is sufficient. This was previously observed in Lake Czechowskie in Northern Poland (Roeser
et al., 2021) and in Diss Mere, England (Boyall et al., 2023). Only at the shallow lakefloor (<12 m) oxic
conditions prevail during the entire year. The position of the transition layer at ~12 m depth is dictated by the
transfer of heat down the water column from the surface water and reflects the thickness of the upper convective
layer which forms the epilimnion. Because the spread of hypoxia starts from the sediment-water interface and
spread upward, it is bounded by the depth of transition layer. The time delay between the vertical oxygen
distribution and temperature dynamics likely reflects the timing and duration of OM decomposition during the
period of stratification at the deepest part of the lake basin and the circulation of lake water that starts from the
bottom and the time required to transport oxygen into the bottom waters.

### 5.2 The transition from oxygenated to hypoxic conditions

The identification of the onset of varve preservation in the sedimentary record is a common tool to date the
onset of hypoxic conditions in lakes because it indicates the absence of bioturbating organisms (e.g., Diaz and
Rosenberg, 1995). However, varve preservation is a binary proxy that is only indicating a depletion of the DO
level to below a certain threshold. To trace possible depletion in DO level prior to varve preservation, more
sensitive geochemical proxies are applied. Geochemical sediment characterization by clustering of XRF data
shows that the varved unit V, and the underlaying unit G are geochemically similar and reflect similar sediment
composition (**Figure 8**). This is also supported by relative variations of endogenic calcite and detrital matter
(Ca/Ti), and relative variations of biogenic silica and detrital matter (Si/Ti) (**Figure 9**). The increase of the
CaCO$_3$ concentrations and Ca/Ti ratio at the transition from unit B to G, reflects an enhanced calcite formation
(**Figure 9**). The increase in the Si/Ti records at the transition to unit G indicates a relative increase of diatom
accumulation that continue into the varved unit V, but is less clear in the shallow core TSK23-SC2 located at a
water depth of less than 12 m. Intervals of increasing calcite and diatom deposition are characterized by cluster

5 in shallow-water cores and cluster 4 in deep-water cores. Together, the rising deposition of calcite and diatom frustules indicate changes in sedimentation prior to the onset of varve preservation, while DO level were still sufficient for the existing of bioturbating organisms.

Variations in Fe/Mn ratios are another sensitive proxy because oxidation of organic matter at the sediment-water interface and upper sediment is coupled to the reduction of Mn or Fe oxides. Decreasing Fe/Mn ratios at the transition from Unit B to G observed in all cores below 12 m water depth shows that redox processes are initiated at this depositional boundary (Figure 9). This suggests that the redox conditions at the sediment-water interface already changed prior to the onset of varve preservation. The bacterial oxidation of organic matter and

reduction of Mn and Fe at the sediment-water interface are associated to changes in the DO level (Liu et al., 2022; Melton et al., 2014; Thamdrup, 2000). Relative constant Fe/Mn ratios in core TSK23-SC2 located above the oxycline (<12 m) does not show a decrease like in cores from locations with presently low DO levels and thus confirms the sensitivity of this proxy to oxygen conditions at the sediment-water interface. This is also supported by the upcore reduction in oxygen-sensitive chironomid abundance and the decrease in profundal

chironomids already at the transition between Units B and G in core TSK15-K5. $\delta^{13}C_{org}$ profile of three cores (TSK11-K1, 62 m; TSK15-K5, 27.3 m; TSK23-SC2, 11.3 m) support the DO depletion in the deep part of the lake, while the shallower lakefloor remained oxygenated. $\delta^{13}C_{org}$ in the deep and intermediate cores show more negative values upcore, approaching $\delta^{13}C_{org}$ of recent organic materials from sediment traps, which experienced minor degradation (supplementary material Figure S4). Because TSK has no riverine input it is

unlikely that changes in the sources of external OM will significantly impact the $\delta^{13}C_{org}$ values. This is also validated by a TOC/TN ratio of ~10 along the Holocene section, which is typical for lacustrine primary productivity rather than terrestrial OM (Drager et al., 2019). Moreover, variations in $\delta^{13}C_{org}$ along the Holocene section are linked to changes between varved and non-varved intervals. $\delta^{13}C_{org}$ values of around - 32‰ characterized all laminated intervals and around -28‰ the non-laminated intervals. The fact that this is a

constant range during the Holocene likely indicate the homogeneous source of OM in the lake over time with a constant $\delta^{13}C_{org}$ value of the primary OM. Thus, we argue that those more negative $\delta^{13}C_{org}$ values are explained by less selective degradation and point to increasing organic matter preservation at the deep and intermediate lakefloor (Meyers, 1997; Spiker and Hatcher, 1987; Wynn, 2007). This indicates the onset of depletion in the DO level prior to the onset of varve preservation at the deep lakefloor, at the boundary between

Units B and G. $\delta^{13}C_{org}$ values in the shallowest core are less negative and constant along the core (**Figure 9**), reflecting oxygenating conditions for the last two centuries. This agrees with observation of DO dynamics in the lake, which show that the upper part of the water column, the top ~12 m, is oxygenated the whole year (**Figure 11A**).

The transition into hypoxic conditions is driven by the eutrophication of the lake as previously suggested by

Kienel et al. (2013) and Dräger et al. (2019). The spread of hypoxia in lakes during the last three centuries was shown to be a combination of changing climate conditions (Jane et al., 2021; Jenny et al., 2014) and human-induced eutrophication (Jenny et al., 2016a) in many temperate lakes. The TSK region is an agricultural landscape area since the last centuries which explains an increased nutrient input into the lake from industrial

fertilizers and sewage. In consequence, the lake internal productivity of the lake increased as reflected by the
TOC increase and the onset of diatom blooms during the deposition of the varved unit. Similar timing of
human-induced hypoxia was reported from other lakes in south Baltic lowlands (Poraj-Górska et al., 2021). The
contribution of climate warming, which enhance water column stratification and reduce lake circulation might
be an additional driver for hypoxia spread as suggested globally (Jane et al., 2021). Sediment focusing was
suggested to bias the primary Fe/Mn signal, challenging the use of Fe/Mn as a proxy for oxygen conditions
(Naeher et al., 2013). In the case of the TSK record, the top varved unit is characterized by frequent fluctuations
of the Fe/Mn ratio, correspond to the number of years, based on the varve counting. This indicate that the
seasonal hypoxia controlled by the lake circulation is well recorded in the sedimentary record of the deep
lakefloor (**Figure 9**). Thus, we conclude that the influence of sediment focusing is negligible, with no substantial
impact on changes in the measured proxies that reflect the depletion of the DO in the lake.

The abundance of remains of chironomid larvae in two cores, from the deep and shallow parts of the lake basin,
supports our hypothesis regarding DO depletion over time. Chironomids are oxygen-sensitive organisms
(Brodersen et al., 2004; Heinis and Davids, 1993; Perret-Gentil et al., 2024; Ursenbacher et al., 2020) that can
live in lacustrine deep-water environments. Of the deep-water chironomid groups identified in the sediments of
TSK, *M. radialis*-type is typically considered the most sensitive to eutrophication and associated oxygen
decrease. Habitats with pronounced hypoxia over extended periods of time are usually characterized by an
absence of deep-water chironomid larvae, low concentrations of chironomid remains and low absolute
abundances and percentages of profundal chironomid groups in subfossil chironomid assemblages (Brodersen
and Quinlan, 2006; Ursenbacher et al., 2020; Van Hardenbroek et al., 2011). Chironomid remains in core
TSK15-K5 from intermediate water depth support the two-step DO depletion in the lake as high concentrations
and influx values of total and profundal chironomids were observed in Unit B, lower abundances in the Unit G
and negligible abundance in Unit V (**Figure 10**). This decrease in DO is also confirmed by the distinct reduction
in *M. radialis*-type, the dominant deep-water taxon in the core (**Figure 10**). The overall low concentrations,
influx values and percentages of profundal chironomids in the deep-water core TSK18-SC4 may indicate that
during the entire studied interval the deep-water core was characterized by lower DO levels compared to the
intermediate core. This indicates that relatively low DO levels in the depocenter of the lake prevailed even
during the deposition of unit B, which is considered to reflect the most oxygenated conditions. However, the DO
level at the time was insufficient for varve preservation and only a further decrease in the DO resulted in the
preservation of varves in the depocenter from the base of Unit V onwards.

In summary, the transition from units B to G marks a change in the geochemical composition of the sediments
driven by initial DO depletion at the lakefloor prior to the onset of varve preservation.  Therefore, the onset of
varve preservation reflects a threshold in DO level and duration of hypoxia required for varve preservation. The
time difference between the onset of DO depletion and the onset of varve preservation provides an estimate of
the sensitivity of the lake system to environmental change.

### 5.3 Time shift between the onset of oxygen depletion and varve preservation

The time the onset of geochemical changes (B-G boundary) and the onset of varve preservation (G-V boundary) is determined through the Askja-1875 cryptotephra at the transition between Units B and G and the varve-dated onset of annual laminations at the different locations and ranges from ~15 yrs in the depocenter (62 m) to ~80-100 yrs at intermediate water depths (27-35 m) (**Figure 12**). This difference implies that the cessation of bioturbation is more rapid in the depocenter compared to intermediate water depths. The faster sediment
response at the depocenter to depleting DO levels can be explained by the following mechanism: (i) The depocenter experienced higher TOC flux, due to sediment focusing and thus enhanced OM decomposition and that drives DO depletion. (ii) At intermediate water depths, DO levels sufficiently low to allow varve preservation were reached later because water column mixing and circulation have more influence on DO levels than on the deep lakefloor. Because seasonal hypoxia starts at lakefloor and spread upward, at intermediate
water depths the duration of hypoxia is shorter and thus also less severe.

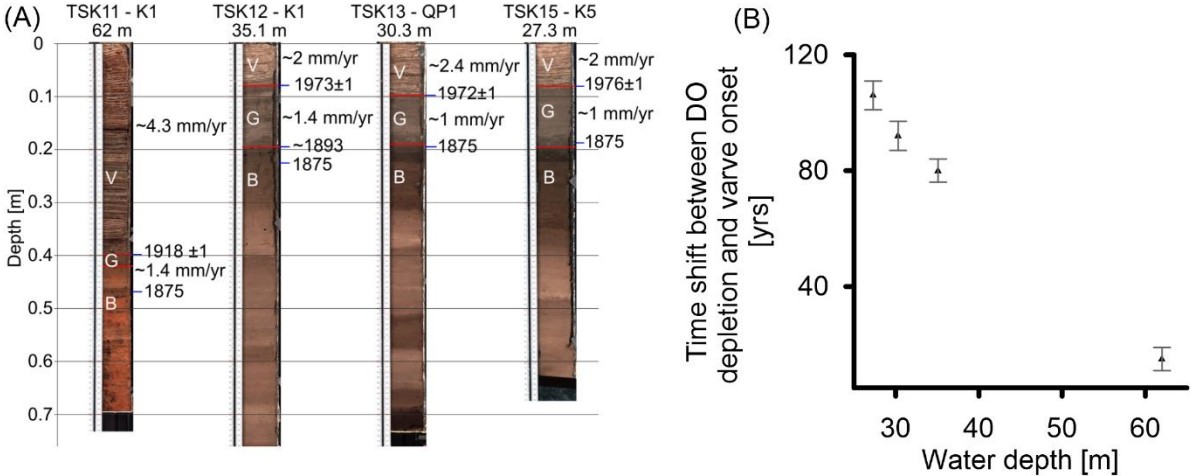

**Figure 12. (A) Images of the selected cores and their chronologic anchors. (B) The time interval between the onset of DO depletion (base of Unit G) and the onset of varve preservation (base of Unit V).**

### 5.4 Basin scale spread of hypoxia

To discuss the basin-wide spread of hypoxia, we use the onset of varve preservation as a proxy for hypoxia because this is best visible in the core and comparable with other studies based on this indicator (e.g., Jenny et al., 2016b), keeping in mind, however, that DO depletion started a few decades earlier. It took ~80 yr from crossing the DO threshold at 62 m water depth (CE 1918±1) until it reached 16 m water depth in CE 1997±1; the shallowest lakefloor depth from which a core with a varved top was taken (**Figure 6**). A similar rate of
hypoxia spread in two phases is observed in two morphometric variables, (i) the lakefloor area, and (ii) the water volume (**Figure 13**). In the first phase, from 1918 and 1972, the rate in which the lakefloor becomes hypoxic was $1 - 3.5 \cdot 10^3 \ m^2/yr$, and the rate of water volume that becomes hypoxic was $10 - 35 \cdot 10^3 \ m^3/yr$. In the second phase between 1972 and 1997 these rates increased substantially reaching a rate of $10 - 14 \cdot 10^3 \ m^2/yr$ for the spread of hypoxia on the lakefloor, and $138 - 164 \cdot 10^3 \ m^3/yr$ of water volume

that became hypoxic. Moreover, between 1918 and 1997, the proportion of the lakefloor area that experienced

hypoxic conditions increased to 52% of the total lakefloor (**Figure 13C**, brown dots), while the water volume that

became hypoxic increased only to 36% of the total water volume (**Figure 13C**, blue dots). This yields an

increasing difference between the percentage of hypoxic lakefloor area and water volume hypoxia over time as

at any water depth, the proportion of the lakefloor that is hypoxic is larger than the proportion of the water

volume. The acceleration in the rate of hypoxia spread in TSK around CE 1972 is controlled by the shape of the

basin and occurred when the level of hypoxia reached a water depth of ~30 m, equal to the slope break (**Figure**

**1**). At that point, the rate in which the lakefloor and the water volume became hypoxic increased by almost an

order of magnitude.

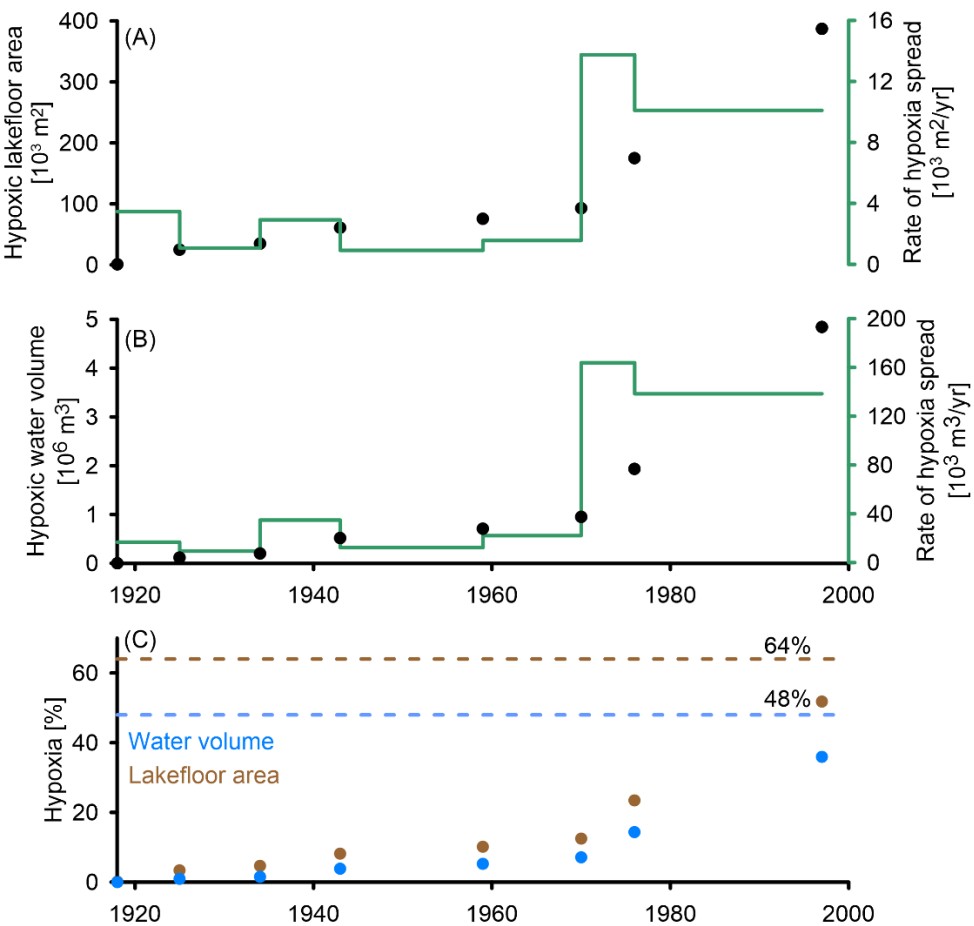

**Figure 13. The spread of hypoxia in the lake (black) and its rate (green) over time. (A) The rate in which the lakefloor**

**becomes hypoxic. (B) The rate in which the lake's water volume becomes hypoxic. (C) Percentage of hypoxic water**

**volume (blue dots) and lakefloor area (brown dots) from the total. The dashed lines indicate the maximal extent of**

**hypoxia under present-day conditions, when the seasonal oxycline is at ~12 m depth.**

## 6    Conclusions

The depletion of DO in the TSK during the past two centuries is recorded by lithological changes and

geochemical and biological proxies allowing to identify the presence of several threshold processes.

1. Preservation of varves marks a depletion of DO level to below a threshold of $[O_2] < 5$ mg/l for five months and $[O_2] < 2$ mg/l for two months under seasonal hypoxic conditions. This provides evidence that the presence of varves not necessarily requires meromictic conditions (permanent stratification).

2. According to changes in sediment geochemistry oxygen levels initially decreased around CE 1875 and continued for another four to five decades until at the depocenter of the lake a threshold in oxygen depletion was crossed allowing varve to be preserved.

3. Hypoxic conditions sufficient to allow varve preservation spread upwards in the lake basin during ~80 years from the deepest part of the lake up to a water depth of 16 m. The upper limit of hypoxic conditions is defined

by the oxycline, roughly reflecting the thermocline, above which permanent oxygenated conditions prevail and no varves are present.

4. The basin morphology controls the spread of hypoxia which occurred in two phases. An acceleration occurred during the 1970s when the level of hypoxia reached the slope break at ~30 m.

5. The biogeochemical cycles in the lake reflected in calcite, diatom production and TOC deposition intensified

along with the development of hypoxic conditions.

**Data availability**

All raw data can be found at the GFZ data archive.

**Author contribution**

IS – conceptualization, data curation, investigation, funding acquisition, visualization, writing - original draft.

RT - data curation, formal analysis, visualization, software, writing - review & editing. SP - data curation, writing - review & editing. BS - data curation, writing - review & editing. MA - data curation. RK - data curation. OH - data curation, visualization, writing - review & editing. SB - data curation, visualization, writing - review & editing. AB – conceptualization, investigation, funding acquisition, Project administration, writing - review & editing.

**Competing interests**

The authors declare that they have no conflict of interest.

**7 Acknowledgements**

Brian Brademann and Karla Wurz are thanked for the assistance in field and lab work. The authors thank two anonymous referees for their careful review. IS was supported by a post-doctoral fellowship from the Alexander

Von Humboldt foundation. The monitoring equipment used in this study was funded by the Terrestrial Environmental Observatory Infrastructure initiative of the Helmholtz Association (TERENO Observatory NE Germany). It is further a contribution to the Helmholtz climate initiative REKLIM (Regional Climate Change and Humans) 'Research Theme 3 Extreme events across temporal and spatial scales'.

**Uncategorized References**

Anderson, R. Y., and Dean, W. E., 1988, Lacustrine varve formation through time:
           palaeogeography, Palaeoclimatology, palaeoecology, v. 62, no. `1-4, p. 215-235.
       Arthur, M. A., and Dean, W. E., 1998, Organic-matter production and preservation and
           evolution of anoxia in the Holocene Black Sea: Paleoceanography, v. 13, no. 4, p.
           395-411.
Benner, R., Fogel, M. L., Sprague, E. K., and Hodson, R. E., 1987, Depletion of 13C in lignin
           and its implications for stable carbon isotope studies: Nature, v. 329, no. 6141, p. 708-
           710.
       Bertrand, S., Tjallingii, R., Kylander, M. E., Wilhelm, B., Roberts, S. J., Arnaud, F., Brown,
           E., and Bindler, R., 2024, Inorganic geochemistry of lake sediments: A review of
analytical techniques and guidelines for data interpretation: Earth Science Reviews, v.
           249, no. 104639.
       Blockley, S. P., Pyne-O'Donnell, S. D., Lowe, J. J., Matthews, I. P., Stone, A., Pollard, A.
           M., Turney, C. S. M., and Molyneux, E. G., 2005, A new and less destructive
           laboratory procedure for the physical separation of distal glass tephra shards from
sediments.: Quaternary Science Reviews, v. 24, no. 16-17, p. 1952-1960.
       Boyall, L., Valcárcel, J. I., Harding, P., Hernández, A., and Martin-Puertas, C., 2023,
           Disentangling the environmental signals recorded in Holocene calcite varves based on
           modern lake observations and annual sedimentary processes in Diss Mere, England:
           Journal of Paleolimnology.
Brauer, A., and Casanova, J., 2001, Chronology and depositional processes of the laminated
           sediment record from Lac d'Annecy, French Alps: Journal of Paleolimnology, v. 25,
           p. 163–177.
       Brauer, A., Schwab, M. J., Brademann, B., Pinkerneil, S., and Theuerkauf, M., 2019, Tiefer
           See – a key site for lake sediment research in NE Germany: DEUQUA Special
Publications, v. 2, p. 89–93.
       Brodersen, K. P., Pedersen, O., Lindegaard, C., and Hamburger, K., 2004, Chironomids
           (Diptera) and oxy-regulatory capacity: an experimental approach to paleolimnological
           interpretation: Limnology and Oceanography, v. 49, no. 5, p. 1549-1559.
       Brodersen, K. P., and Quinlan, R., 2006, Midges as palaeoindicators of lake productivity,
eutrophication and hypolimnetic oxygen: Quaternary Science Reviews, v. 25, no. 15-
           16, p. 1995-2012.
       Brooks, S. J., Langdon, P. G., and Heiri, O., 2007, The identification and use of Palaearctic
           Chironomidae larvae in palaeoecology.
       Buatois, L. A., Renaut, R. W., Owen, R. B., Behrensmeyer, A. K., and Scott, J. J., 2020,
Animal bioturbation preserved in Pleistocene magadiite at Lake Magadi, Kenya Rift
           Valley, and its implications for the depositional environment of bedded magadiite:
           Scientific Reports, v. 10, no. 1.
       Davies, B. R., 1976, The dispersal of Chironomidae larvae: a review: Journal of the
           Entomological Society of Southern Africa, v. 39, no. 1, p. 39-62.
Dean, W. E., and Gorham, E., 1998, Magnitude and significance of carbon burial in lakes,
           reservoirs, and peatlands: Geology, v. 26, no. 6, p. 535-538.
       Diaz, R. J., and Rosenberg, R., 1995, Marine benthic hypoxia: A review of its ecological
           effects and the behavioural response of benthic macrofauna: Oceanography and
           marine biology: An annual review, v. 33, p. 245-303.
-, 2008, Spreading dead zones and consequences for marine ecosystems: science, v. 321, no.
           5891, p. 926-929.

Dräger, N., Plessen, B., Kienel, U., Słowinski, M., Ramisch, A., Tjallingii, R., Pinkerneil, S., and Brauer, A., 2019, Hypolimnetic oxygen conditions influence varve preservation and d13C of sediment organic matter in Lake Tiefer See, NE Germany: Journal of Paleolimnology, v. 62, p. 181–194-181–194.

Dräger, N., Theuerkauf, M., Szeroczyńska, K., Wulf, S., Tjallingii, R., Plessen, B., Kienel, U., and Brauer, A., 2017, Varve microfacies and varve preservation record of climate change and human impact for the last 6000 years at Lake Tiefer See (NE Germany): The Holocene, v. 27, no. 3, p. 450– 464-450– 464.

Evans, G., Augustinus, P., Gadd, P., Zawadzki, A., and Ditchfield, A., 2019, A multi-proxy μ-XRF inferred lake sediment record of environmental change spanning the last ca. 2230 years from Lake Kanono, Northland, New Zealand: Quaternary Science Reviews, v. 225.

Friedrich, J., Janssen, F., Aleynik, D., Bange, H. W., Boltacheva, N., Çagatay, M. N., and Wenzhöfer, F., 2014, Investigating hypoxia in aquatic environments: diverse approaches to addressing a complex phenomenon: Biogeosciences, v. 11, no. 4, p. 1215-1259.

He, W., You, L., Chen, M., Tuo, Y., Liao, N., Wang, H., and Li, J., 2023, Varied sediment archive of Fe and Mn contents under changing reservoir mixing patterns, oxygenation regimes, and runoff inputs: Ecological Indicators, v. 147, no. 109967.

Heinis, F., and Davids, C., 1993, Factors governing the spatial and temporal distribution of chironomid larvae in the Maarsseveen lakes with special emphasis on the role of oxygen conditions: Netherland Journal of Aquatic Ecology, v. 27, p. 21-34.

Hunt, J. B., and Hill, P. G., 1996, An inter-laboratory comparison of the electron probe microanalysis of glass geochemistry: Quaternary International, v. 34, p. 229-241.

Jane, S. F., Hansen, G. J., Kraemer, B. M., Leavitt, P. R., Mincer, J. L., North, R. L., and Rose, K. C., 2021, Widespread deoxygenation of temperate lakes: Nature, v. 594, no. 7861, p. 66-70.

Jankowski, T., Livingstone, D. M., Bührer, H., Forster, R., and Niederhauser, P., 2006, Consequences of the 2003 European heat wave for lake temperature profiles, thermal stability, and hypolimnetic oxygen depletion: Implications for a warmer world: Limnology and Oceanography, v. 51, no. 2, p. 815-819.

Jenny, J.-P., Normandeau, A., Francus, P., Ecaterina, Z., Taranue, I., Eavese, G., Lapointea, F., Jautzy, J., Ojala, A., E. K. , Dorioz, J.-M., Schimmelmannk, A., and Zolitschkal, B., 2016a, Urban point sources of nutrients were the leading cause for the historical spread of hypoxia across European lakes: Proceedings of the National Academy of Sciences, v. 13, no. 45, p. 12655–12660.

Jenny, J. P., Arnaud, F., Alric, B., Dorioz, J. M., Sabatier, P., Meybeck, M., and Perga, M. E., 2014, Inherited hypoxia: A new challenge for reoligotrophicated lakes under global warming: Global Biogeochemical Cycles, v. 28, no. 12, p. 1413-1423.

Jenny, J. P., Arnaud, F., Dorioz, J. M., Giguet Covex, C., Frossard, V., Sabatier, P., Millet, L., Reyss, J. L., Tachikawa, K., Bard, E., Pignol, C., Soufi, F., Romeyer, O., and Perga, M. E., 2013, A spatiotemporal investigation of varved sediments highlights the dynamics of hypolimnetic hypoxia in a large hard-water lake over the last 150 years: Limnology and Oceanography, v. 58, no. 4, p. 1395-1408.

Jenny, J. P., Francus, P., Normandeau, A., Lapointe, F., Perga, M. E., Ojala, A., Schimmelmann, A., and Zolitschka, B., 2016b, Global spread of hypoxia in freshwater ecosystems during the last three centuries is caused by rising local human pressure: Global Change Biology, v. 22, no. 4, p. 1481-1489.

Jochum, K. P., Stoll, B., Herwig, K., Willbold, M., Hofmann, A. W., Amini, M., Aarburg, S., Abouchami, W., Hellebrand, E., Mocek, B., Raczek, I., Stracke, A., Alard, O., Bouman, C., Becker, S., Dücking, M., Brätz, H., Klemd, R., Bruin, D., Canil, D., Cornell, D., Hoog, C., Dalpé, C., Danyushevsky, L., Eisenhauer, A., Gao, Y., Snow, J. E., Groschopf, N., Günther, D., Latkoczy, C., Guillong, M., Hauri, E. H., Höfer, H.
E., Lahaye, Y., Horz, K., Jacob, D. E., Kasemann, S. A., Kent, A. J. R., Ludwig, T., Zack, T., Mason, P. R. D., Meixner, A., Rosner, M., Misawa, K., Nash, B. P., Pfänder, J., Premo, W. R., Sun, W. D., Tiepolo, M., Vannucci, R., Vennemann, T., Wayne, D., and Woodhead, J. D., 2006, MPI-DING reference glasses for in situ microanalysis: New reference values for element concentrations and isotope ratios:
Geochemistry, Geophysics, Geosystems, v. 7, no. 2.

Kastowski, M., Hinderer, M., and Vecsei, A., 2011, Long-term carbon burial in European lakes: Analysis and estimate: Global Biogeochemical Cycles, v. 25, no. 3.

Kelts, K., and Hsü, K. J., 1978, Freshwater Carbonate Sedimentation, *in* Lerman, A., ed., Freshwater carbonate sedimentation. In Lakes: chemistry, geology, physics: New
York, NY, Springer New York, p. 295-323.

Kienel, U., Dulski, P., Ott, F., Lorenz, S., and Brauer, A., 2013, Recently induced anoxia leading to the preservation of seasonal laminae in two NE-German lakes: Journal of Paleolimnology, v. 50, no. 4, p. 535-544.

Kienel, U., Kirillin, G., Brademann, B., Plessen, B., Lampe, R., and Brauer, A., 2017, Effects
of spring warming and mixing duration on diatom deposition in deep Tiefer See, NE Germany: Journal of Paleolimnology, v. 57, no. 1, p. 37-49.

Lane, C. S., Cullen, V. L., White, D., Bramham-Law, C. W. F., and Smith, V. C., 2014, Cryptotephra as a dating and correlation tool in archaeology: Journal of Archaeological Science, v. 42, p. 42-50.

Lehmann, M. F., Bernasconi, S. M., Barbieri, A., and McKenzie, J. A., 2002, Preservation of organic matter and alteration of its carbon and nitrogen isotope composition during simulated and in situ early sedimentary diagenesis: Geochimica et Cosmochimica Acta, v. 66, no. 20, p. 3573-3584.

Liu, J., Chen, Q., Yang, Y., Wei, H., Laipan, M., Zhu, R., He, H., and Hochella Jr, M. F.,
2022, Coupled redox cycling of Fe and Mn in the environment: The complex interplay of solution species with Fe-and Mn-(oxyhydr) oxide crystallization and transformation: Earth-Science Reviews, v. 232, no. 104105.

Loizeau, J. L., Span, D., Coppee, V., and Dominik, J., 2001, Evolution of the trophic state of Lake Annecy (eastern France) since the last glaciation as indicated by iron,
manganese and phosphorus speciation: Journal of Paleolimnology, v. 25, p. 205-214.

Makri, S., Wienhues, G., Bigalke, M., Gilli, A., Rey, F., Tinner, W., and Grosjean, M., 2021, Variations of sedimentary Fe and Mn fractions under changing lake mixing regimes, oxygenation and land surface processes during Late-glacial and Holocene times: Science of the total environment, v. 755, no. 143418.

Meire, L., Soetaert, K. E. R., and Meysman, F. J. R., 2013, Impact of global change on coastal oxygen dynamics and risk of hypoxia: Biogeosciences, v. 10, no. 4, p. 2633-2653.

Melton, E. D., Swanner, E. D., Behrens, S., Schmidt, C., and Kappler, A., 2014, The interplay of microbially mediated and abiotic reactions in the biogeochemical Fe
cycle: Nature Reviews Microbiology, v. 12, no. 12, p. 797-808.

Mendonça, R., Müller, R. A., Clow, D., Verpoorter, C., Raymond, P., Tranvik, L. J., and
Sobek, S., 2017, Organic carbon burial in global lakes and reservoirs: Nature
communications, v. 8, no. 1, p. 1694.

Meyers, P. A., 1997, Organic geochemical proxies of paleoceanographic, paleolimnologic,
and paleoclimatic processes: Organic geochemistry, v. 27, no. 5-6, p. 213-250.

Mollenhauer, G., and Eglinton, T. I., 2007, Diagenetic and sedimentological controls on the
composition of organic matter preserved in California Borderland Basin sediments:
Limnology and Oceanography, v. 52, no. 2, p. 558-576.

Mulholland, P. J., and Elwood, J. W., 1982, The role of lake and reservoir sediments as sinks
in the perturbed global carbon cycle: Tellus, v. 34, no. 5, p. 490-499.

Naeher, S., Gilli, A., North, R. P., Hamann, Y., and Schubert, C. J., 2013, Tracing bottom
water oxygenation with sedimentary Mn/Fe ratios in Lake Zurich, Switzerland:
Chemical Geology, v. 352, p. 125-133.

Njiru, M., Nyamweya, C., Gichuki, J., Mugidde, R., Mkumbo, O., and Witte, F., 2012,
Increase in anoxia in Lake Victoria and its effects on the fishery, *in* Padilla, P., ed.,
Anoxia, p. 99-128.

Nürnberg, G. K., 2004, Quantified Hypoxia and Anoxia in Lakes and Reservoirs: The
Scientific World Journal, v. 4, p. 42–54.

O'Reilly, C. M., Sharma, S., Gray, D. K., Hampton, S. E., Read, J. S., Rowley, R. J.,
Schneider, P., Lenters, J. D., McIntyre, P. B., Kraemer, B. M., Weyhenmeyer, G. A.,
Straile, D., Dong, B., Adrian, R., Allan, M. G., Anneville, O., Arvola, L., Austin, J.,
Bailey, J. L., Baron, J. S., Brookes, J. D., De Eyto, E., Dokulil, M. T., Hamilton, D.
P., Havens, K., Hetherington, A. L., Higgins, S. N., Hook, S., Izmest'Eva, L. R.,
Joehnk, K. D., Kangur, K., Kasprzak, P., Kumagai, M., Kuusisto, E., Leshkevich, G.,
Livingstone, D. M., MacIntyre, S., May, L., Melack, J. M., Mueller-Navarra, D. C.,
Naumenko, M., Noges, P., Noges, T., North, R. P., Plisnier, P. D., Rigosi, A.,
Rimmer, A., Rogora, M., Rudstam, L. G., Rusak, J. A., Salmaso, N., Samal, N. R.,
Schindler, D. E., Schladow, S. G., Schmid, M., Schmidt, S. R., Silow, E., Soylu, M.
E., Teubner, K., Verburg, P., Voutilainen, A., Watkinson, A., Williamson, C. E., and
Zhang, G., 2015, Rapid and highly variable warming of lake surface waters around
the globe: Geophysical Research Letters, v. 42, no. 24, p. 10773-10781.

Ojala, A. E., Saarinen, T., and Salonen, V. P., 2000, Preconditions for the formation of
annually laminated lake sediments in southern and central Finland: Boreal
Environment Research, v. 5, no. 3, p. 243-255.

Perret-Gentil, N., Rey, F., Gobet, E., Tinner, W., and Heiri, O., 2024, Human impact leads to
unexpected oligotrophication and deepwater oxygen increase in a Swiss mountain
lake: The Holocene, v. 34, no. 2, p. 189–201.

Poraj-Górska, A. I., Bonk, A., Żarczyński, M., Kinder, M., and Tylmann, W., 2021, Varved
lake sediments as indicators of recent cultural eutrophication and hypolimnetic
hypoxia in lakes: Anthropocene, v. 36, no. 100311.

Roeser, P., Drager, N., Dariusz, B., Ott, F., Pinkerneil, S., Gierszewski, P., Lindemann, C.,
Plessen, B., Brademann, B., Kaszubski, M., Schwab, M. J., Slowinski, M.,
Blaszkiewicz, W., and Brower, A., 2021, Advances in understanding calcite varve
formation : new insights from a dual lake monitoring approach in the southern Baltic
lowlands: Boreas.

Saether, O. A., 1979, Chironomid communities as water quality indicators: Ecography, v. 2,
no. 2, p. 65-74.

Sanchini, A., Szidat, S., Tylmann, W., Vogel, H., Wacnik, A., and Grosjean, M., 2020, A Holocene high-resolution record of aquatic productivity, seasonal anoxia and meromixis from varved sediments of Lake Łazduny, North-Eastern Poland: insight from a novel multi-proxy approach: Journal of Quaternary Science, v. 35, no. 8, p. 1070-1080.

Schaffner, L. C., Jonsson, P., Diaz, R. J., Rosenberg, R., and Gapcynski, P., Benthic communities and bioturbation history of estuarine and coastal systems: effects of hypoxia and anoxia, *in* Proceedings Marine Coastal Eutrophication1992, Elsevier, p. 1001-1016.

Shatkay, M., Anati, D. A., and Gat, J. R., 1993, Dissolved oxygen in the Dead Sea - seasonal changes during the holomictic stage: International Journal of Salt Lake Research, v. 2, no. 95, p. 93-110.

Sobek, S., Durisch-Kaiser, E., Zurbrügg, R., Wongfun, N., Wessels, M., Pasche, N., and Wehrli, B., 2009, Organic carbon burial efficiency in lake sediments controlled by oxygen exposure time and sediment source: Limnology and Oceanography, v. 54, no. 6, p. 2243-2254.

Sorrel, P., Jacq, K., Van Exem, A., Escarguel, G., Dietre, B., Debret, M., and Oberhänsli, H., 2021, Evidence for centennial-scale Mid-Holocene episodes of hypolimnetic anoxia in a high-altitude lake system from central Tian Shan (Kyrgyzstan): Quaternary Science Reviews, v. 252, no. 106748.

Spiker, E. C., and Hatcher, P. G., 1987, The effects of early diagenesis on the chemical and stable carbon isotopic composition of wood: Geochimica et Cosmochimica Acta, v. 51, no. 6, p. 1385-1391.

Steinsberger, T., Schmid, M., Wüest, A., Schwefel, R., Wehrli, B., and Müller, B., 2017, Organic carbon mass accumulation rate regulates the flux of reduced substances from the sediments of deep lakes: Biogeosciences, v. 14, no. 13, p. 3275-3285.

Straile, D., Jöhnk, K., and Rossknecht, H., 2003, Complex effects of winter warming on the physicochemical characteristics of a deep lake: Limnology and Oceanography, v. 48, no. 4, p. 1432-1438.

Teranes, J. L., and Bernasconi, S. M., 2005, Factors controlling $\delta^{13}C$ values of sedimentary carbon in hypertrophic Baldeggersee, Switzerland, and implications for interpreting isotope excursions in lake sedimentary records: Limnology and Oceanography, v. 50, no. 3, p. 914-922.

Thamdrup, B., 2000, Bacterial manganese and iron reduction in aquatic sediments, Advances in microbial ecology: Boston, Springer, p. 41-84.

Theuerkauf, M., Blume, T., Brauer, A., Dräger, N., Feldens, P., Kaiser, K., and Schult, M., 2022, Holocene lake-level evolution of Lake Tiefer See, NE Germany, caused by climate and land cover changes: Boreas, v. 51, no. 2, p. 299-316.

Tjallingii, R., Röhl, U., Kölling, M., and Bickert, T., 2007, Influence of the water content on X-ray fluorescence core-scanning measurements in soft marine sediments: Geochemistry, Geophysics, Geosystems, v. 8, no. 2.

Tranvik, L. J., Downing, J. A., Cotner, J. B., Loiselle, S. A., Striegl, R. G., Ballatore, T. J., and Weyhenmeyer, G. A., 2009, Lakes and reservoirs as regulators of carbon cycling and climate: Limnology and oceanography, v. 54, no. 6, p. 2298-2314.

Tylmann, W., Szpakowska, K., Ohlendorf, C., Woszczyk, M., and Zolitschka, B., 2012, Conditions for deposition of annually laminated sediments in small meromictic lakes: A case study of Lake Suminko (Northern Poland): Journal of Paleolimnology, v. 47, no. 1, p. 55-70.

Tyson, R. V., and Pearson, T. H., 1991, Modern and ancient continental shelf anoxia: an overview: Geological Society, London, Special Publications, v. 58, no. 1, p. 1-24.

Ursenbacher, S., Stötter, T., and Heiri, O., 2020, Chitinous aquatic invertebrate assemblages in Quaternary lake sediments as indicators of past deepwater oxygen concentration: Quaternary science reviews, v. 231, no. 106203.

Van Hardenbroek, M., Heiri, O., Wilhelm, M. F., and Lotter, A. F., 2011, How representative are subfossil assemblages of Chironomidae and common benthic invertebrates for the living fauna of Lake De Waay, the Netherlands?: Aquatic Sciences, v. 73, p. 247-259.

Vaquer-Sunyer, R., and Duarte, C. M., 2008, Thresholds of hypoxia for marine biodiversity: Proceedings of the National Academy of Sciences, v. 105, no. 40, p. 15452-15457.

Weltje, G. J., Bloemsma, M. R., Tjallingii, R., Heslop, D., Röhl, U., and Croudace, I. W., 2015, Prediction of geochemical composition from XRF core scanner data: a new multivariate approach including automatic selection of calibration samples and quantification of uncertainties, Micro-XRF Studies of Sediment Cores: Applications of a non-destructive tool for the environmental sciences, p. 507-534.

Wetzel, R. G., 2001, Limnology: lake and river ecosystems, Elsevier.

Wiederholm, T., 1983, Chironomidae of the holarctic region. Keys and diagnoses. Part 1: larva, Ent Scand Suppl, Volume 19, p. 1-457.

Wulf, S., Dräger, N., Ott, F., Serb, J., Appelt, O., Guðmundsdóttir, E., Van den Bogaard, C., Slowinski, M., Blaszkiewicz, M., and Brauer, A., 2016, Holocene tephrostratigraphy of varved sediment records from Lakes Tiefer See (NE Germany) and Czechowskie (N Poland): Quaternary Science Reviews, v. 132, p. 1-14.

Wynn, J. G., 2007, Carbon isotope fractionation during decomposition of organic matter in soils and paleosols: Implications for paleoecological interpretations of paleosols: Palaeogeography, Palaeoclimatology, Palaeoecology, v. 251, p. 437–448.

Zander, P. D., Żarczyński, M., Vogel, H., Tylmann, W., Wacnik, A., Sanchini, A., and Grosjean, M., 2021, A high-resolution record of Holocene primary productivity and water-column mixing from the varved sediments of Lake Żabińskie, Poland: Science of the total environment, v. 755, no. 143713.

Żarczyński, M., Wacnik, A., and Tylmann, W., 2019, Tracing lake mixing and oxygenation regime using the Fe/Mn ratio in varved sediments: 2000 year-long record of human-induced changes from Lake Żabińskie (NE Poland): Science of the Total Environment, v. 657, p. 585-596.

Zolitschka, B., Francus, P., Ojala, A. E. K., and Schimmelmann, A., 2015, Varves in lake sediments - a review: Quaternary Science Reviews, v. 117, p. 1-41.