# Peer review of "Tracing rate and extent of human induced hypoxia during the last 200 years in the mesotrophic lake Tiefer See (NE Germany)"

_EGUsphere, 2024_

## Author Comment (AC1)

**Authors reply in bold red**

Referee #1

The authors tackle an important, ongoing, and rapidly progressing environmental issue of lacustrine hypoxia. I appreciate the less common approach where multiple cores from the lake bottom transect are used. This kind of approach, rightfully so, requires more work than typical investigations based on the depocenter core. Multiple cores provide an important outlook on the water column structure as well as the resulting sedimentation processes. However, there are major issues that should be resolved before the final version is accepted.

**We thank the referee or his careful and supportive review**

1) The microfacies analysis methodology is described in detail, however, the results are negligible or almost absent, mostly being a (justified) reference to previous works. Scanned thin sections were used to correlate and construct composite profiles, whereas slabs were used for µXRF mapping. While all that is fine and expected, early on the text leaves an impression that the actual microfacies analysis is a part of the reported investigations. The beginning of the conclusion says "The depletion of DO in the TSK during the past two centuries is recorded by varve microfacies and geochemical and biological proxies (…)". In my opinion it should be much more general, and say that overall, it is recorded in the lithological variability. Parts of the manuscript talking about varve quality and diffuse character, in my opinion, do not justify the reference to microfacies.

**The statement in the conclusion section was revised from "varve microfacies" to "varve chronology" as well as in the method section 3.2.**

2) The manuscript lacks an explanation of the mechanisms responsible for the lack of oxygen and overall change of the lake mixing, hypolimnion (de)oxygenation, and so on. I'd expect some more discussion on the possible influence of productivity changes, given that the biogenic silica proxy is introduced. This is briefly mentioned in line 480, as possibly increased TOC loadings. But where does it come from, and why? Compared to the results discussion seems to be not developed enough and lacks references.

**The discussion regarding drivers for hypoxia was elaborated (see lines 528-538). Indeed, the hypoxia is driven mainly by eutrophication of the lake and increased primary productivity and decomposition of the organic materials.**

Of the three major goals found in the introduction, goals I and III are not fully met or backed up by discussion. The discussion lacks also in terms of references. Not to look to far, there are studies on lacustrine hypoxia from Germany, Poland, Switzerland, some even using multiple cores.

**We think that goals 1 and 2 were achieved in this study. Goal 1 seeks to reconstruct the spatiotemporal spread of hypoxia. This is achieved by discussion section 5.4 (and figures 6 and 13), which presents in detail the rate of hypoxia spread in the lake. Goal 2 seeks to decipher the rate of hypoxia spread in lake TSK and the response of the sedimentary system to depleted DO levels. This is achieved by Section 5.2, which shows the impact of hypoxia spread on different properties of the sediments and the transition into varved sedimentation (Figures 7-10). In addition, section 5.3 (figure 12) evaluates the time span between the first evidence for oxygen depletion and the onset of hypoxic conditions as indicated from the onset of varve preservation and complement the effort of identifying the rate of change in the lake. Several references were added in this aspect.**

**Objective 3 was modified to fit the text.**

3) Generally, I'd expect the limnological characterization to be presented before the sedimentary material because it gives the proper context and describes the entire setting where further investigations are carried out. I suggest that the Authors consider changing the paper structure, because now it reads backwards, with the sedimentary response being presented before the potential processes responsible for the formation of the sedimentary signal.

**We added more background information in the introduction (lines 117-121).**

**Regarding the results and discussion, because this is a sedimentologic study that uses diverse proxies to trace changes in the lake and that the monitoring data is only used to complement the sedimentological perspective, we first present the characterization of the cores, and then the limnological part. However, the discussion starts with the limnological framework, so the sedimentological interpretation is read with the limnological context.**

4) Paragraph 3.3.1 has no information on the counting strategy. How was the varve counting uncertainty developed? How many counts were done, and how many people were involved? One person or multiple?

**Information was added in section 3.3.1 (lines 168-174). Varve counting was done by two people, each one of them counted the varves multiple times. The uncertainty is related to the number of varves that was counted between two consecutive marker layers. Because varve preservation is very good, and the cores were easily correlated, the uncertainty is small.**

5) Extensive and temporally spread coring of the TSK leads to inevitable numerous core codes and IDs that are easy to mix up. I'd suggest that every time a core is discussed or reported its depth is given in the parentheses to give proper, in-line context. It is a little bit straining to scroll back and forth to remember the depth.

**Implemented**

6) I have mixed feelings about B-G-V unit naming. V for varved is facies-related, whereas B for basal and G for "grayish homogeneous gyttja" is a mix of concepts and descriptors. It makes relating the units a little more complex.

**The naming is based on visual impression. V for varved, G for gray, B for brown. See Figure 2 caption.**

7) Minor suggestion, some words seem to be unnecessarily written with hyphens. We do not write paleo-limnology or paleo-climate so why write paleo-redox? Furthermore, I noticed instances of UK spelling – e.g. palaeoecological in line 86, while paleoredox is written in the US. Please carefully check capitalization, too. It is inconsistent in the text and figures.

**Corrected through the text.**

Specific and minor comments

50 and further: reasoning in this sentence and later in the manuscript needs to be clarified. While the absence of DO is a condition for varve deposition, it is not a matter of straightforward reversal in the case of massive sediments, thus it is not a binary system. The absence of evidence is not evidence of absence.

**We agree and revised the text.**

59: something is missing in this sentence.

**Rephrased**

60: while the meaning is similar, I cannot agree that XRF is a method for oxygen reconstruction. Elemental data is the proxy here, XRF is just a means of obtaining it. Should someone run sequential extraction of for example Fe and Mn reasoning would be the same. XRF is changing and enhancing how this data is obtained. The next sentence is the right choice. Consider introducing elemental data first and then explaining the importance of XRF.

**We agree and clarified it in the text (lines 66-68).**

90: Just Tiefer See.

**Implemented.**

120: Indeed, there are two higher-resolution proxy datasets, but the enumeration goes to three and includes chironomids. Correct the logic here.

**Corrected.**

132: Please, explain why vertical.

**To allow for particles at the top of the core to settle and to not disturb the layers. It is written in the beginning of the sentence.**

Paragraph 3.3 is unacceptably short.

**Expanded. Note that subsections 3.3.1 and 3.3.2 detail the 3.3 chronology section.**

175: provide manufacturer.

**Manufacturers of both machines used are mentione in lines 193-194 and 200 respectively**

176: do you mean a dwell time of 4 seconds per point?

**Yes, with new deterors, higher current (mA), and the Rh X-Ray source short measurement times can be strongly reduced**

178: provide details on the foil. Furthermore, intensities were probably acquired for many more elements, these elements that are listed, were selected afterward.

**Unfortunately, there are no details on the foil provided by the producer. Actually, non of the publications on ITRAX core scanning data provides information on this.**

**Indeed, default XRF spectra contain many more elements. We applied a quality test based on 3-fold replicate measurements and only present the elements with a relative standart deviation of less than 25 %. We have changed the text accordingly (Lines 205-209).**

181: this is a rather definitive and bold statement, that log-ratio transformed data is free of mentioned problems.

**Indeed, log ratios are not free of matrix effects. Element intensities are linear functions of log-ratios of concentrations and provide a linear quatification of the relative matrix effects as described in Weltje & Tjallingii 2008. We changed the text accordingly (lines 211-215).**

Entire paragraph on XRF: I cannot find information on how clrs were calculated. Was it XELERATE? Anyway, please provide a reference.

**We rephrase this part of the text and added the refrerences that describe log-ratio transformations and the Xelerate software packge (lines 214-216).**

192: that's the first time that GFZ full name appears, which seems odd, given that most probably XRF and EPMA and other analyses were carried out there as well.

**It is mentioned now in the other method sections as well.**

193: combusted?

**Corrected**

211: Please identify the microscope (chironomid).

223: "scalar field of the water column" read a little bit like an unnecessary complication which brings no substance here.

**Rephrased.**

240: "…varves defined by Roeser et al. (2021) as a triplet of diatom-rich…"

260: suffices to say it was obtained in 2018, no need to do the math.

**Deleted.**

303: Si as an authigenic mineral doesn't seem fitting, it is either diatomaceous opal, which is a little bit different than authigenic calcite, for example. If it is not biogenic silica, then silicates are detrital.

**Section 4.4.1 on the xrf cluster analysis was revised and detailed. The meaning is that Si is mostly originated from the diatom frustules (biogenic silica) with a minor detrital component. This is why although the diatoms and calcite layer appear together in the varved unit, Si is not pointing to the same direction as the Ca. The fact that Si do not go with Ti and K means that only minor part of the Si is of detrital origin. As mentioned in the introduction, the lake experienced spring diatom blooms and only minor detrital flux (no river input, no dust), thus the Si signal is much endogenic.**

355: I don't think that the data fully supports the statement, that the transition is not visible in the Fe/Mn. In my opinion, there is a transition in Fe/Mn, which is visible once the scale is adjusted. It might not be as sharp and isochronous, but it is clear.

**The Fe/Mn scale in the three cores is the same in order to compare between the change in the Fe/Mn ratio. There might be a slight change also in the shallow core, but not substantial change and of course not a stepwise change as in the intermediate and deep cores. I can adjust the scale to highlight the changes in the Fe/Mn along the shallow core, but I think this will be misleading because small changes in the shallow core will look similar to much bigger changes in the deeper cores.**

Paragraph 4.4.1: Information on the algorithm or distance choice should be provided in the methods section, as well as any data pretreatment or transformation and used software. Please move it accordingly. Why 5 clusters?

**The interpretation of the statistical results has now been revised in §4.4.1 to better explain our interpretation of the element proxies and our choice of 5 clusters. We address our choice of 5 clusters in line 352-355.**

**The choice of 5 clusters was done because we wanted to trace variations along deep/shallow gradient and between laminated/non-laminated textures. We applied the cluster analysis with up to 10 clusters and it did not give any benefit.**

I'm not sure if I follow. Does the PCA biplot, figure 7, represent all the data points across multiple cores and, therefore different deposition settings? **Yes, see figure 7 caption**. From the PCA biplot and vector angles it seems that Fe is a little bit closer to the detrital components than redox-sensitive. I do not have a proper solution, but I'd appreciate it if the author put some extra consideration into the geochemical clusters versus lithological units. Reporting that clusters of the same lithology are different numbers makes it challenging for the reader.

**The description of each cluster was expanded and a lithological description was added near each cluster number (lines 361-367). The point is that the cores are divided into three units (B, G, V) based on their textures and colors, but the geochemical data (xrf) coupled with the statistical analyses show that actually units G and V are geochemically similar and that the same unit in different cores (shallow vs. deep cores) can actually be geochemically distinct. These are fundamental observations in this research and they provide an additional layer of understanding of the depositional processes in the lake.**

**The Fe represents a mixture between redox conditions and detrital origin and this is why its angle is in between the Mn (redox) and Ti (detrital) vectors.**

347: It is a very clear example that data visualization is heavily influencing the way results are reported. Simply cutting the picture and stretching it along the x axis shows, that this record is not as flat as the Authors suggest.

**This comment looks similar the one related to line 355. See reply there.**

360–363: In my opinion, this fits more into the Discussion than the results report.

**Deleted from the results section**

385: This is somewhat expected that the anoxia develops till the last moment, whereas thermal stratification peaks before it.

**Agreed. It is not highlighted as a new finding just as a description of the data.**

401: Is this a typical way to write units in BG journal? Figures use forward-slash, the same for ratios in text.

**BG journal demand using SI or SI derived units. Thus, oxygen conc. Can be expressed in mole or g. mg/l along with mmol/l is commonly used in such cases. The slash were corrected.**

406: I'd expect some references here, showing that in mono and dimictic lakes, this is a generally found phenomenon, where short pulses of oxygenation do not lead to the destruction of varves.

**Added.**

408: How does thermocline development stop oxycline migration? Consider providing some more critical explanation. 409: What does it mean that the circulations start from the bottom? Generally, I have trouble with these few sentences around line 409/410.

**Those sentences were rewritten and this is clarified now (lines 473-478). Not the circulation starts from the bottom, the hypoxic conditions start from the bottom and spreads upward.**

418: Well, this is expected, assuming that the material deposited has the same source and genesis. Homogeneous sediments are mixed and eventually undergo different early diagenesis processes given changes in the oxygenation, however, these are not expected to be drastically different, especially at times when the lake oscillates around the varve-formation threshold.

**This is obvious after the fact and we cannot make this assumption because the lake undergone significant change in sedimentation due to eutrophication and the question is when does the lake enters into the seasonal sedimentation regime. Unit B for example is geochemically different than units G and V. The question is if the composition of the sediments is different also between Unit G in the middle to the top Unit V. The cluster analysis shows that they are similar geochemically and that the big change in the composition of the sediment occurs at the transition from Unit B to Unit G.**

428: Shallow cores are typically less enriched in this respect, given that the lake bottom is depocenter and accumulates more matter in general and in principle.

**This is true that there is a sediment focusing in the lake and that the deposition rate is higher at the depocenter. However, diatoms and calcite form at the upper water column (<12 m), thus it is not clear at all that when we look at ratios like Ca/Ti and Si/Ti as proxies for the composition of the sediments, we will see depth differences.**

434: As before, I cannot fully agree that Fe/Mn is that constant, even though it fits the narrative.

**We think the stepwise change in the Fe/Mn ratio in the intermediate and deep cores that is correlated with the transition from Unit B to Unit G reflect fundamentally different behavior there than in the shallowest core.**

480: Introduction of the DO controlling mechanism is much appreciated, but needs some more discussion.

**The discussion was expanded (lines 528-538).**

495 and later: If it is not somehow normalized (how?) then it is expected that lower depths equal higher area affected, just by definition of lake morphometry. I cannot say that these numbers are conclusive and support the aim.

**This is correct, and maybe expected. We agree that the numbers themselves are not so important, however it may still be important to provide at rates in absolute numbers and of course in percentage.**

Figures and tables

Figure 1: A) Baltic Sea; B) Rearrange the depth bar in the legend, so it follows the topography. No need for high/low, only numbers. Bring to front monitoring platform icon. Graticule axes on the map have no units. Generally, this introductory paragraph could be rewritten for clarity.

**The figure was edited accordingly**

Figure 4: Could you possibly identify marker layers on the entire core picture?
**It is difficult to do that precisely because of the challenge to identify the marker layers on the entire core layer. This is why we used the overlapping thin sections. It is even more impossible to correlate the different cores using the entire core rather than using the thin sections.**

Figure 6: there is no axis label on axis x.

**Added**

Figure 9: While I respect the idea that all the axes between the profiles have the same range which I supposed to help with comparison it is also artificially punctuating points made by authors. Relative changes recorded in one sedimentary environment are not that easy compared to the other deposition settings. Therefore, for example, panel B looks artificially flat when compared to the other cores. This leads to some unwarranted result descriptions in the text. I strongly suggest that the axes and ranges are selected by proxy and by the panel to accurately showcase the Second issue, while compared to the high-resolution XRF data results of the other analyses are discrete, these dots can be still connected to better show the tendencies. Consider putting at least a few selected time markers on the plots, which are now depth-resolved and, thus not that easily comparable. Finally, the figure caption does not match the plot sequence.

**The point of this figure is to show proxies for DO indicate oxygen depletion at deep water vs. no depletion in the shallow water (represented by the core from 11 m). Thus, the scale of the x axis for each proxy must be similar to highlight those changes.**

Figure 10: While the general idea of calculating fluxes as a mean of comparison is right, and makes absolute sense, these comparisons make sense in the text. Here line on panel B seems to be misleading. Furthermore, sediments from shallower parts of the lake, and residing on the slopes by definition have lower fluxes. Relative changes within a given depth interval should be studied more, rather than made into rigid comparison with the depocenter values.

**The comparison here is not between the cores but focus on relative changes upcore which follow the process of oxygen depletion. Of course, that the deeper core will show higher flux because of recycling and this is why the different chironomid groups were assembled in different ways. For example, the profundal species (deep water species; black line) exclude recycled species.**

Figure 11: Panel A, has an average oxycline depth, the same goes for panel B and thermocline. Panel B could use a more gradual scale (more steps). Please consider that for people with deuteranopia color scale used for panel B might be impossible to differentiate adequately.

**Figures were edited**

---

## Author Comment (AC2)

Authors reply in bold red

Referee #2

**General comments to the Author:**

Sirota et al. develop a comprehensive analysis of the dynamics of bottom lake oxygen conditions in Lake Tiefer See over the past 200 years. This is a significant and timely topic for lakes worldwide, as long-term quantitative extents, causes, and consequences of hypoxia on e.g. the carbon burial efficiency, are still not fully understood. The authors rely on a spatial approach based on the analysis of multiple cores retrieved from different water depths, coupled with a multi-proxy approach, including chironomids, TOC, Mn:Fe ratios, δ13C, and varve preservation.

The sedimentological and geochemical analysis are sounds and exhaustive. The authors have invested considerable effort in providing a detailed description and analysis of the multiple cores.

Congruency in the multi proxies' responses strengthen conclusions on past hypoxia dynamics reconstruction in Lake Tiefer. Findings reveal a progressive alteration of the oxygenation conditions in the lake over the past two centuries, evidenced by an increased in the depth of hypoxia within the water column from 62 to 16 m. Furthermore, the authors highlight that the progressive spread of hypoxia has been associated with a series of transitions in the lake's geochemical and biological paleorecords, indicating varying sensitivities to hypoxia spread. This quantitative reconstruction thus sheds new light on the progression rhythms of hypoxia and provides valuable information on markers that can reconstruct hypoxia.

**We thank the referee for his careful review and constructive comments. Below you find our detailed reply.**

**Below, Authors will find three main comments on the manuscript:**

The objectives and raised questions are clearly outlined. However, limited attention is given to the drivers of hypoxia, despite it being stated as one of the study's objectives. The authors should either expand their efforts in this regard or consider removing the objective related to this aspect.

**Objective 3 was revised and the discussion in this regard was elaborated in lines 528-538.**

Robust dating is essential for high-resolution environmental reconstructions. Although this study presents three different dating methods, the connection to previous work on sedimentary dating using absolute methods should be more clearly stated. Specifically, radionuclide varve chronology has already been validated by 12 radiocarbon dates of organic macroremains (Brauer et al. 2019 https://doi.org/10.5194/deuquasp-2-89-2019, 2019). This earlier study reinforces confidence in the varve dating approach proposed here and should be more explicitly highlighted before using varves as dating approach (i.e. method and results section).

**Agreed, see edition in methods section 3.3.1. Note that previous dating was done only for one depocentral core, and in this study, we expand varve counting and tephrochronology to additional cores to achieve a better spatial coverage.**

The authors note that a decline in organic carbon degradation is observed as D13C decreases. This decrease could also be linked to a shift in carbon sources, towards e.g. a greater contribution from in-lake productivity. This would align with the concomitent increased fluxes in diatom frustules and biogenic calcite precipitation observed in the record. I am curious if the authors may have

underestimated the role of primary production associated with cultural eutrophication in the initiation of hypoxia and the subsequent preservation of varves (L 445).

**Indeed, this is a main issue when dealing with organic matter content in lacustrine records. This is now clearer along the text in the introduction (lines 60-63 and lines 124-125) and discussion (lines 513-520). Our conclusion that a decline in OM degradation is observed along the cores is based on two arguments: 1. In our analyses, as well as along the Holocene record of the lake the TOC/TN ratio is ~10, typical to the ratio of lacustrine primary productivity. For comparison, the TOC/TN ratio of terrestrial OM is ~40, thus this lake does not experience a substantial OM external contribution (Drager et al., 2019). This agrees with the fact that there is no river inflow to the lake. 2. The D13C of the OM along the Holocene record ranges from -28 during non-varved intervals (oxygenated conditions) to -32 during varved intervals (hypoxic conditions). The fact that varved intervals which reflect hypoxic conditions are also characterized with D13C that characterize less degraded OM at different stages during the Holocene indicate that the sources of OM (lacustrine primary productivity) in the lake stay the same over time. Thus, the starting point of the D13C value of the OM is constant, while the actual value that is measured along the cores depends on the degree of OM degradation that shift those values.**

**Dräger, N., Plessen, B., Kienel, U., Słowiński, M., Ramisch, A., Tjallingii, R., ... & Brauer, A. (2019). Hypolimnetic oxygen conditions influence varve preservation and δ 13 C of sediment organic matter in Lake Tiefer See, NE Germany. Journal of paleolimnology, 62, 181-194.**

**Specific comments**

I'm uncertain about 'Lake Tiefer See,' but as it is presented, it seems redundant to me. Perhaps just 'Tiefer Lake' or 'Tiefer See' would work?

**We use the lake Tiefer See or TSK (Tiefer See klocksin) throughout the text. This is the correct form.**

Abstract L13 : prefer "a detailed quantification of hypoxia spread on centennial timescales remained largely unquantified" to "a detailed quantification of hypoxia spread remained largely unquantified". Note that recent synthesis on DO trends based on linological records provide quantification of hypoxia spread (e.g. Jane et al,. Nature 2021)

**Implemented. This is an important paper in the topic and it is now referenced in the manuscript. Jane et al. (2021) do not focus on a single lake, but they integrate limnological data (temperature and oxygen measurements) from hundreds of lakes and show trends in their properties. Moreover, they do not study hypoxia from the sedimentological aspect.**

L15: "associated with" instead of "associated by"?

**Corrected.**

L17: "in 1997±1" instead of "at 1997±1"

**Corrected.**

L17: "and reached a lake-floor depth of 16 m at 1997±1" : is that the maximum extent record? Clarify

**Clarified in the section 5.4 of the discussion. The core from 16 m depth is the shallowest core where varves, as a proxy for hypoxia were found at the top. At shallower lakefloor depth, ~11 m,**

**no varves were found, thus hypoxia did not prevail there. From this reason, we can follow the spread of hypoxia based on this sedimentological evidence.**

L20: "threshold for hypoxia" Threshold for what? Varve onset? Death of macro benthic lives?

**The onset of varve preservation marks the stop of bioturbation activity due to the death of bioturbating benthic life. In this case this happens because of the DO decline. Here we quantify the DO value (threshold), i.e., intensity and duration, in which below it the varve preservation is achieved.**

**This is now better explained in the text (lines 20-21).**

L22: "depletion in DO started several decades prior to the varve preservation" Do you mean DO decline? drop below a certain DO threshold ?

**Varve preservation is achieved as oxygen level drop to below a certain threshold which we quantify in this paper. However, the start of oxygen decline occurs earlier as indicated by the geochemical proxies. Clarified in Line 22.**

L23: "accomplished" maybe change to something else, e.g. "started"

**Revised**

L34: "natural and anthropogenic processes". Not sure you want to use "processes", maybe "pressure" instead

**Implemented.**

L36: "decreased lake circulation" Do you mean water residence time? Winter mixing? Else?

**Mixing.**

L39-40: this is a bit of a shortcut. Burial of OM compensate part (not totally) of the C emissions to the atmosphere, in that sense those fluxes are important but it is a bit to exaggerated to say that it contributes to substantial contribution to the global carbon fixation (it is currently ~<0.1Pg/a, far below the anthropic emissions), maybe on geological timescale, not on annual basis…

**Agreed, I added the time scale in Line 40. we cannot claim the increase in organic C fixation in lake sediments compensate the anthropogenic emission. It is just referring to the change in organic C fixation in lake sediments under changing oxygenated conditions on short geological time scales (100s-1000s years). Note, that in the text we state that this is compared with oceanic organic C fixation.**

L44: Be more specific on your motivation

**Revised in lines 45-46**

L50: "[…] while non-laminated intervals reflect DO level sufficient for the existence of bioturbation organisms." This is only accurate for lakes with a seasonally contrasted sedimentation regime.

**This is written in general, and I think that any type of fine lamination (sub-mm to mm), no matter what is time scale represented by the lamination, will emphasize the absence of bioturbation. Moreover, seasonal lamination is highly common in lakes.**

L54 : "Limited oxygen availability for laminated sediment intervals" not clear… Low oxygen conditions in bottom waters leading to varve preservation…?

**The sentence was rephrased**

L56: "OM degradation results in selective degradation of organic compounds with a more negative $\delta 13C$ composition and explains the more negative $\delta Corg$ 13 values measured in laminated sediments" Could you verify this statement. $\delta^{13}C$ values can result from various factors, such as the sources of organic matter, metabolic transformations, and biological fractionation... Worth stating why $\delta^{13}C$ can be specifically used in Lake Tiefer as OM degradation indicator.

**Indeed. This is now better explained in the discussion 5.2 section (lines 513-520). See detailed response to this issue in your third main comment above.**

L76 : why mesotrophic lakes only? The method can work also in oligo or eu-trophic lakes...

**This is true and was deleted.**

L120: Authors state 2 independent proxies of hypoxia while 3 proxies are presented... Please clarify this.

**Revised.**

L135: I suggest to add a statement to explain why the thin sections are made for. This is explain in the next sections but a statement there may help the reading.

**Revised (line 150-151).**

L396 : DO is not controlled by the hydrodynamics but also by oxygen consumption rates (i.e. biological and chemical DO demand).

**Sentence was rephrased.**

Important finding to me is that the thermocline depth limits the spread of hypoxia in Lake Tiefer. Should be more emphasised in the MS

**Elaborated in lines 473-478.**

L459 : Authors indicate a « two-step DO depletion in the lake ". Decrease in DO can be progressive, triggering step by step impacts on the biogeochemical records. The steps are more relative to the impact of DO decrease on the sediment records than on DO trends itself... ?

**This is true. The decrease in DO level is probably progressive, but this is indicated by two transitions in the sediments driven by changes in oxygen level. We can guess that between the start in DO decline (Units B-G transition) until it achieves the threshold for varve preservation, which is marked by the onset of varve preservation, the oxygen declines progressively. However, as a sedimentological study we think that we should describe the processes from the sedimentological point of view and from the observations that it provides.**

---

## Author Response (AR2)

General and major comments

Generally, the Authors addressed most of the reviewers' comments or justified their decision not to change parts of the text. The manuscript is substantially improved and, after addressing minimal concerns, should be published in Biogeosciences.

I understand the reasoning behind the author's choice of the unified scales of the plots and their narrative on different rates of change registered in different lake zones, especially given their further explanation, which has some merit. Even though I accept this choice, I must articulate that I can't entirely agree with it. But this might be a matter of perspective and different principles.

We thank the referee for his careful review.

Specific and minor comments

65: The sentence about the X-ray ends abruptly. Otherwise, I appreciate the improved text here.

The sentence was rephrased.

70: these references could use some fundamental works, like Naeher 2013, Engstrom 1985…

I thank the reviewer for his suggestions that were added as references.

494: riverine…

Implemented.

518: A little more care in reasoning about the role of focusing (or lack thereof) - especially geochemical focusing with recurring stratification/mixing events - is needed here. Specifically, Fe and Mn are potentially enriched every time the lake goes through the turnover/stratification cycle. It fits the story of strengthened hypoxia, stratification, and eutrophication.

Implemented. The discussion on sediment focusing was expanded (see lines 518-523).